# Structure and rational engineering of the PglX methyltransferase and specificity factor for BREX phage defence

Sam C. Went [1], David M. Picton[1], Richard D. Morgan[2], Andrew Nelson[3], Aisling Brady [4], Giuseppina Mariano [5], David T. F. Dryden[1], Darren L. Smith [3], Nicolas Wenner[4], Jay C. D. Hinton [4] & Tim R. Blower [1] ✉

Bacteria have evolved a broad range of systems that provide defence against their viral predators, bacteriophages. Bacteriophage Exclusion (BREX) systems recognise and methylate 6 bp non-palindromic motifs within the host genome, and prevent replication of non-methylated phage DNA that encodes these same motifs. How BREX recognises cognate motifs has not been fully understood. In this study we characterise BREX from pathogenic *Salmonella* and present X-ray crystallographic structures of the conserved BREX protein, PglX. The PglX N-terminal domain encodes the methyltransferase, whereas the C-terminal domain is for motif recognition. We also present the structure of PglX bound to the phage-derived DNA mimic, Ocr, an inhibitor of BREX activity. Our analyses propose modes for DNA-binding by PglX and indicate that both methyltransferase activity and defence require larger BREX complexes. Through rational engineering of PglX we broaden both the range of phages targeted, and the host motif sequences that are methylated by BREX. Our data demonstrate that PglX is used to recognise specific DNA sequences for BREX activity, contributing to motif recognition for both phage defence and host methylation.

Bacteria have evolved a diverse range of defences to protect from bacteriophages (phages) and mobile genetic elements[1,2]. Classic examples of host defence mechanisms include restriction-modification (RM)[3], abortive infection[4,5] and CRISPR-*cas*[6]. Genes encoding these systems tend to co-localise into "defence islands"[7]. Analysis of defence islands using a "guilt-by-association" approach[8] or functional selection[9], has resulted in significant expansion of predicted and validated defence systems[8,9], including Bacteriophage Exclusion (BREX)[10], CBASS[11], BstA[12], retrons[13], viperins[14], pycsar[15] and PARIS[16]. Whilst the combinations of phage defence systems encoded in any island can differ, there is evidence that conserved regulatory systems, such as the BrxR family, control

defence expression perhaps mediating robust defence against a broad spectrum of invaders[17–19].

BREX genes are found in 10% of bacterial and archaeal genomes[10]. BREX is related to Phage Growth Limitation (Pgl)[20] and was first identified through analysis of genes neighbouring *pglZ*, performed to locate likely defence genes[10]. Together with *gmrS/gmrD*, which encode a Type IV restriction enzyme, BREX genes form one of the most common defence island pairings[7,21]. We have recently demonstrated that a defence island encoded on a multidrug-resistant plasmid of *Escherichia fergusonii* provides complementary phage defence using BREX and a GmrSD homologue, BrxU[22]. There are six BREX sub-types, and type I BREX contains six genes; *brxA*, *brxB*, *brxC*, *pglX*, *pglZ* and

[1]Department of Biosciences, Durham University, South Road, Durham, UK. [2]New England Biolabs, 240 County Road, Ipswich, MA, USA. [3]Faculty of Health and Life Sciences, Northumbria University, Newcastle Upon Tyne, UK. [4]Institute of Infection, Veterinary and Ecological Sciences, University of Liverpool, Liverpool, UK. [5]Department of Microbial Sciences, Faculty of Health and Medical Sciences, University of Surrey, Guildford, UK. ✉e-mail: timothy.blower@durham.ac.uk

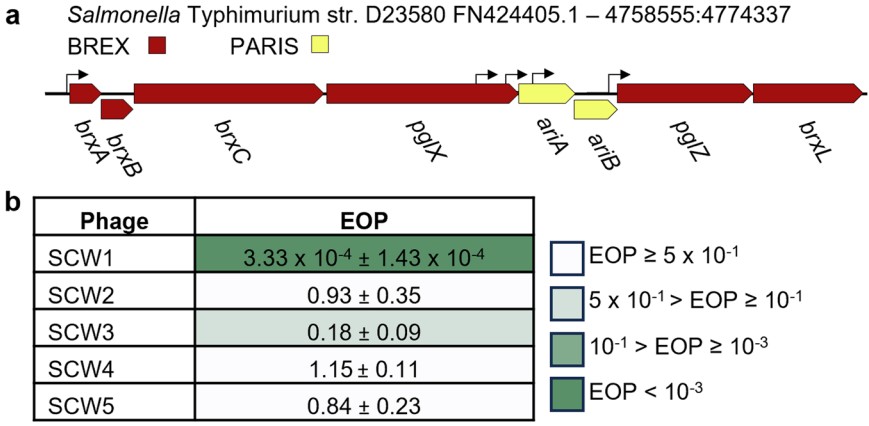

**a** *Salmonella* Typhimurium str. D23580 FN424405.1 – 4758555:4774337

**Fig. 1 | *Salmonella* BREX provides phage defence against environmentally isolated *Salmonella* phages. a** Schematic of the 15.7 kb *Salmonella* BREX phage defence island. Promoters are denoted by arrows. **b** Efficiency of Plating (EOP) for *Salmonella* phages tested on *Salmonella* D23850Δφ against a control of *Salmonella* D23580ΔφΔBREX. Values are mean EOPs from triplicate data, shown with standard deviation.

*brxL*[10]. BrxA is a DNA-binding protein[23], and BrxL is a DNA-stimulated AAA+ ATPase[24]. PglX has sequence and structural homology to methyltransferases and is hypothesised to methylate non-palindromic 6 bp sequences (BREX motifs) on the N6 adenine at the fifth position of the motif[10,22,25], allowing discrimination between self and non-self DNA. Interestingly, it has been shown that Ocr from phage T7, a protein that mimics dsDNA[26], can inhibit BREX activity through binding to PglX[27]. Whilst reminiscent of RM systems, the mechanism of BREX activity remains unclear.

The *stySA* locus from *Salmonella enterica* serovar Typhimurium[28], (also known as SenLT2III), was recently re-constructed in an attenuated lab strain of *S.* Typhimurium (LT2) and shown to have BREX activity[29]. In 2017, invasive non-typhoidal *Salmonella* (iNTS) disease was responsible for 77,500 deaths globally, of which 66,500 deaths occurred in sub-Saharan Africa[30]. A high proportion of African iNTS cases are caused by *S.* Typhimurium ST313[31,32]. Representative ST313 strain D23580[31] encodes a BREX locus that is closely-related to the LT2 BREX locus (Fig. 1a), comprising a defence island formed from an amalgamation of the type I BREX system and PARIS[16]. The D23580 BREX defence island lacks the additional upstream and regulatory genes observed in the *E. fergusonii* type I BREX defence island[22].

The relative simplicity of the *Salmonella* BREX system and the clinical relevance of the host strain prompted us to test the effects of the D23580 BREX defence island against environmental *Salmonella* phages. The D23580 BREX phage defence island was then characterised through systematic gene deletions in an *E. coli* background, to allow use of the Durham phage collection[33] in identifying the determinants of phage defence and PglX-dependent host methylation. We present the first X-ray crystallographic structural characterisation of PglX. We also present the first X-ray crystallographic structural characterisation of PglX bound by the DNA mimic Ocr. Through rational engineering of PglX it was possible to alter the BREX motif recognised for methylation and phage defence. Our structural and biochemical analyses support PglX being the BREX methyltransferase and suggest modes of DNA-binding through the PglX C-terminal domain. Our data also definitively show PglX determines specificity within BREX phage defence, providing motif recognition for both phage targeting and host methylation.

## Results

### The *Salmonella* D23580 BREX phage defence island provides protection against environmental *Salmonella* phages

The BREX phage defence island from *Salmonella enterica* serovar Typhimurium ST313 strain D23580 (referred to as D23580 from now on) encodes two phage defence systems, type I BREX[10], and PARIS[16], collectively "BREX$_{Sty}$" (Fig. 1a). The SalComD23580 RNA-seq-based gene expression compendium (http:/bioinf.gen.tcd.ie/cgi-bin/sal-com_v2.pl?_HL) shows that the defence island is expressed constitutively at the transcriptional level during exponential growth in LB and minimal media, and within murine macrophages[34]. Differential RNA-seq (dRNA-seq) was used to identify a promoter upstream of *brxA* (*STMMW_44431*) at location 4773879 on the D23580 chromosome, which drives transcription of the BREX-PARIS island[34] (Fig. 1a).

Also known as StySA[28], the ~15.7 kb D23580 BREX$_{Sty}$ phage defence island has two synonymous point mutations in *pglX* compared to the model *S.* Typhimurium ST19 strain LT2. The BREX island has recently been studied in the *S.* Typhimurium-derived strain ER3625. Phage transduction was used to construct ER3625 as a genetic hybrid between *S.* Abony 803 strain and *S.* Typhimurium in the 1960's, and the strain has recently been sequenced[35]. In comparison to D23580, the defective BREX phage defence island of *S.* Typhimurium strain ER3625 had a further 12 point mutations, of which 7 were distributed throughout *pglZ*, and 5 in the 3'-terminal section of *brxC*[29].

The contiguous PARIS defence systems mediate an abortive infection response in the presence of the anti-BREX and anti-restriction protein Ocr[16]. The co-localisation of the PARIS genes *ariAB* within BREX$_{Sty}$ raises the possibility that the BREX and PARIS defences work together in *S.* Typhimurium. As we have previously studied a phage defence island from *Escherichia fergusonii* ATCC 35469 plasmid pEFER that encodes a BREX system co-localised with a GmrSD-family Type IV restriction enzyme, BrxU[22], we were curious to determine how commonly BREX systems co-localise with other defence systems. DELTA-BLAST was first used to identify genomes encoding distant homologues of PglZ. From 12,000 genomes, ~9,000 genomes co-localised *pglZ* and *pglX* and ~7000 encoded a gene or genes between the BREX components. ~5,000 of these intergenic regions did not encode a currently known phage defence system in-between *pglZ* and *pglX*, as defined by either DefenseFinder[36] or PADLOC 2.0[37]. Of the remaining ~2000, 15.98% encoded BrxU homologues, whilst 2.21% encoded PARIS homologues, making them two of the most common defence systems associated with BREX (Supplementary Fig. 1).

Our first aim was to confirm BREX$_{Sty}$ activity in D23580. To assess phage defence in D23580 we needed to isolate *Salmonella* phages. As phages isolated on D23580 wild type (WT) would be inherently resistant to BREX$_{Sty}$, we first used a genetic approach to generate a strain of D23580 that lacked BREX$_{Sty}$. The ST313 strain D23580 encodes 5 prophages that encode their own antiphage systems, including the prophage BTP1-encoded BstA[12]. To reduce interference from other

antiphage systems, we began with the D23580Δφ mutant strain that lacks the five major prophages. The entire BREX$_{Sty}$ defence island, including PARIS, was then removed from D23580Δφ using scar-less λ red recombination (Supplementary Fig. 2)[38], resulting in strain D23850ΔφΔBREX[39].

Sewage effluent was obtained direct from source with the assistance of Northumbrian Water, and together with water taken from the River Wear in Durham, was used for phage enrichment on D23850ΔφΔBREX. A range of plaques were obtained after these enrichments, and five phage lysates were prepared following rounds of purification from visually distinct plaques. Activity of the D23580 BREX defence island was confirmed using EOP assays with the five *Salmonella* phage isolates, testing the ability of the phages to plaque on D23580Δφ, with D23850ΔφΔBREX as the control (Fig. 1b). An EOP value of less than 1 indicates that a phage is less efficient at forming plaques on the test strain compared to the control. Phage SCW1 had a pronounced reduction of EOP, at 3.33 x 10$^{-4}$; phage SCW3 was moderately reduced for plating, with an EOP of 0.18 (Fig. 1b). The remaining three phages appeared unaffected by activity of BREX$_{Sty}$, with EOPS -1 (Fig. 1b). These data confirm that the BREX$_{Sty}$ defence island of D23580Δφ can provide active anti-phage activity in *Salmonella*.

## Impact of *Salmonella* D23580 BREX phage defence island gene deletions on phage defence and methylation

Having investigated the impact of the D23580 BREX phage defence island, BREX$_{Sty}$, in the original *Salmonella* host, we investigated BREX$_{Sty}$ in an *E. coli* background. The motivation for using this heterologous host was to allow direct comparison with the previously characterised BREX phage defence island from *E. fergusonii*[22], and use of our Durham collection of phages[33]. *E. coli* is also a more tractable experimental model for future experiments within this study. BREX$_{Sty}$ was subcloned in sections and then combined into plasmid pGGA by Golden Gate Assembly (GGA)[40], yielding plasmid pBrxXL$_{Sty}$ that contained the entire BREX and PARIS defence island, namely the eight genes from *brxA* to *brxL* as depicted (Fig. 1a), under the control of the native promoters (Supplementary Fig. 3). Plasmid pTRB507 is an equivalent empty vector control. Liquid cultures of *E. coli* DH5α WT, or cultures transformed with either pBrxXL$_{Sty}$ or pTRB507, were infected with Durham phage TB34[33], or lab phage T7 (ATCC BAA-1025-B2) (Fig. 2a–c). Infected control cultures were lysed by both phages; the T7-infected cultures did not recover, whereas the TB34-infected cultures began to grow again at 10-12 hrs post-infection, presumably due to the selection of spontaneous TB34-resistant mutants (Fig. 2a, b). In the presence of pBrxXL$_{Sty}$, however, cultures infected with TB34 grew similarly to uninfected controls, whilst cultures infected with T7 were lysed (Fig. 2c). These findings show that BREX$_{Sty}$ is active in an *E. coli* background, and demonstrates that pBrxXL$_{Sty}$ provides defence against TB34, but not against T7.

To investigate the role of each phage defence gene in protection against TB34 infection, we generated individual deletions of each D23580 BREX/PARIS gene in pBrxXL$_{Sty}$, and a double mutant that lacked both the *ariA* and *ariB* genes of the PARIS system (Supplementary Fig. 3). *E. coli* DH5α cells were transformed with the mutant plasmids and liquid cultures of resulting strains were subsequently infected with TB34 and T7 (Fig. 2d–l). Deletion of *brxA*, *brxB*, *brxC*, *pglX* and *pglZ* abolished defence against TB34 (Fig. 2d–h). Our finding that deletion of *brxL* did not impact protection against TB34 revealed that BrxL is not required for the phage defence activity of BREX$_{Sty}$ against TB34 (Fig. 2i). Deletion of *aria* and *ariB*, either singly or together, also did not alter defence against TB34 (Fig. 2j–l).

Protection from infection by TB34 and T7 was then monitored using the quantitative EOP assay (Fig. 3a). BREX$_{Sty}$ encoded on pBrxXL$_{Sty}$ provided a moderate 100-fold reduction in TB34 plating efficiency and had no appreciable impact on T7 (Fig. 3a). The 100-fold reduction matches the scale of phage defence observed in *Salmonella*

D23580Δφ against *Salmonella* phages (Fig. 1b). Therefore, plasmid pBrxXL$_{Sty}$ and BREX$_{Sty}$ in the natural host chromosome provide a similar level of defence. Consistent with results obtained with liquid cultures, deletion of *brxA*, *brxB*, *brxC*, *pglX* and *pglZ* ablated phage defence in the EOP assay (Fig. 2; Fig. 3a). However, whereas deletion of *brxL* did not appear to impact protection in liquid cultures (Fig. 2i), the EOP measurements revealed 10,000-fold enhancement of defence against TB34 in the absence of *brxL* compared to cells carrying pBrxXL$_{Sty}$ WT (Fig. 3a). Individual deletion of PARIS genes *ariA* and *ariB* caused a 10-fold increase in phage defence, while the double *ariA*, *ariB* deletion had no additional impact (Fig. 3a). Collectively, these data demonstrate that TB34 is targeted by type I BREX in the BREX$_{Sty}$ D23580 BREX defence island, and that unlike the *E. coli* and *Acinetobacter* BREX systems[17,25], BrxL is not necessarily a requirement for phage defence.

The EOP results of TB34 when tested against the *brxL* deletion and *ariA*, *ariB* double deletion strains (Fig. 3a) prompted us to test a wider range of phages. Using the Durham collection of 12 coliphages[33], we re-tested all phages against pBrxXL$_{Sty}$, pBrxXL$_{Sty}$-Δ*brxL* and pBrxXL$_{Sty}$-Δ*ariA*Δ*ariB* (Supplementary Fig. 4a). Phages TB34, Alma, BB1, CS16, Mav and Sipho had 10- to 100-fold reduced EOPs on pBrxXL$_{Sty}$, compared to empty vector controls (Supplementary Fig. 4a). The *brxL* deletion caused a range of impacts. In some cases we observed enhanced defence (TB34, Alma, Sipho), but in other cases there was no difference to an already susceptible phage (BB1, CS16, Mav) (Supplementary Fig. 4a, b). With phage Pau, against which BREX$_{Sty}$ WT had little effect, the *brxL* deletion enhanced defence (Supplementary Fig. 4a). Other phages unaffected by the WT pBrxXL$_{Sty}$ plasmid were also not impacted by pBrxXL$_{Sty}$-Δ*brxL* (Supplementary Fig. 4a). In contrast, the pBrxXL$_{Sty}$-Δ*ariA*Δ*ariB* construct generally produced similar EOP values compared to pBrxXL$_{Sty}$ WT, though there was an approximate ten-fold further reduction in EOP for phages Alma and Sip (Supplementary Fig. 4a), and there was one major difference where the *ariA*, *ariB* double deletion massively reduced the EOP of BB1 compared to pBrxXL$_{Sty}$ WT (Supplementary Fig. 4a). These data show that the PARIS system was itself not active against any tested phage, and that deletion of *brxL* has phage-dependent impacts on defence (Supplementary Fig. 4a). Due to the unexpected results of a *brxL* deletion in the *E. coli* model, a *brxL* deletion was made in *Salmonella* D23580Δφ in order to test the impact in the original host (Supplementary Fig. 4c). Phages SCW1 and SCW3 are BREX sensitive (Fig. 1b), and this was not impacted by *brxL* deletion (Supplementary Fig. 4c). Phage SCW2 does not appear BREX sensitive when challenged with the WT D23580Δφ strain (Fig. 1b), but has moderately enhanced BREX sensitivity with the *brxL* deletion (Supplementary Fig. 4c). Phages SCW4 and SCW5 are insensitive to either strain. These data corroborate our previous data in *E. coli* (Fig. 3a), and demonstrate that (i) BrxL is dispensable in the native host and (ii) deletion of *brxL* can lead to an enhanced BREX activity in the native host.

Having performed systematic analysis of gene deletions on phage defence, we then investigated a second BREX phenotype; DNA methylation. PglX methyltransferases from type I BREX loci generate N6-methylated adenines (N6mA) at the fifth position within 6-bp non-palindromic motif sequences of host DNA[10,22,25]. Study of the *Salmonella* LT2 StySA BREX system identified GATC**A**G as the target motif sequence[29]. We explored the use of the MinION next-generation sequencing system to detect N6mA methylation patterns[41]. Previously, we performed this type of analysis using methylation-deficient *E. coli* ER2796[42] in order to reduce background methylation. However, we were unable to transform strain *E. coli* ER2796 with our pBrxXL$_{Sty}$ constructs, perhaps because the defence island impacted upon bacterial fitness in the absence of methylation. We therefore used *E. coli* DH5α strains, noting that the background G**A**TC methylation might interfere with detection of the proposed GATC**A**G BREX methylation motif. Total genomic DNA was extracted from each strain and

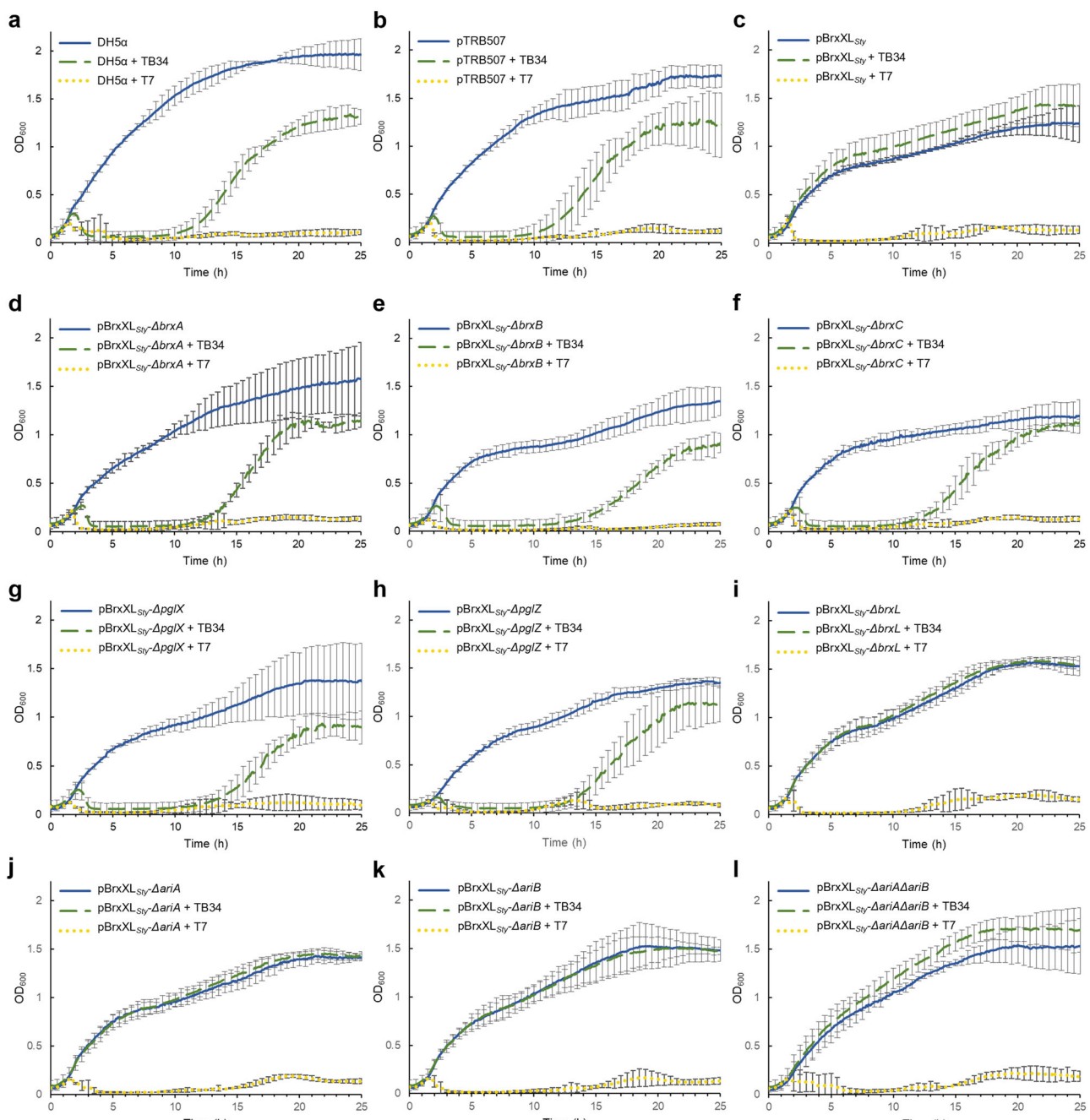

**Fig. 2 | Genes *brxA*, *brxB*, *brxC*, *pglX* and *pglZ* are required for *Salmonella* BREX defence.** Knock-out analysis of the *Salmonella* D23580 BREX phage defence island. **a–l** Growth of *E. coli* DH5α strains harbouring cloned pBrxXL*Sty* WT and mutant plasmids, or pTRB507 control, in the absence or presence of phages TB34 and T7. Data shown are triplicate and error bars represent the standard deviation of the mean.

sequenced by MinION. *E. coli* DH5α pBrxXL*Eferg*, encoding the BREX phage defence island from *E. fergusonii*, was used as an initial positive control to ensure the methylation detection procedure was working. We successfully identified the GCTA**A**T methylation motif (Supplementary Fig. 5a), as previously reported[22]. To confirm the *Salmonella* BREX motif we used a baseline control, wherein the pBrxXL*Sty* WT sample was subjected to whole genome amplification (WGA), which should remove DNA modifications. The WGA sample contained the lowest detectable level of methylated GATC**A**G sequences, 12.87%, whilst pBrxXL*Sty* WT showed GATC**A**G methylation at 78.78% of sites, confirming that D23580 BREX produces N6mA at GATC**A**G sequences (Fig. 3b; and Supplementary Fig. 5b). The *brxA*, *brxB*, *brxC*, *pglX* and *pglZ* mutants showed reduced numbers of GATC**A**G methylation sites

(Fig. 3b), indicating that all five gene products are required for methylation. This finding is consistent with results involving the *Acinetobacter* BREX[17], but differs from those obtained with *E. coli* BREX; the *E. coli* *brxA* was not required for methylation in conditions of arabinose-induced BREX expression[25]. In *S.* Typhimurium BREX, deletion of *brxL* did not reduce methylation (Fig. 3b) and the *ariA*, *ariB* and double mutants showed approximately WT levels of methylation (Fig. 3b).

The observed changes in methylation levels identified the genetic requirements for BREX-mediated methylation. However, the data did not agree with quantitative data on BREX methylation obtained previously from Pacific Biosciences (PacBio) sequencing[22]. To perform a direct comparison, we used the same 12 strains to generate samples for

**a**

| Construct | EOP TB34 | EOP T7 |
|---|---|---|
| pBrxXL$_{Sty}$ | $2.86 \times 10^{-2} \pm 1.54 \times 10^{-2}$ | $1.36 \pm 3.03 \times 10^{-1}$ |
| pBrxXL$_{Sty}$-$\Delta brxA$ | $4.72 \times 10^{-1} \pm 3.02 \times 10^{-1}$ | $7.33 \times 10^{-1} \pm 8.06 \times 10^{-2}$ |
| pBrxXL$_{Sty}$-$\Delta brxB$ | $4.42 \times 10^{-1} \pm 3.85 \times 10^{-1}$ | $8.27 \times 10^{-1} \pm 4.23 \times 10^{-1}$ |
| pBrxXL$_{Sty}$-$\Delta brxC$ | $4.86 \times 10^{-1} \pm 3.71 \times 10^{-1}$ | $1.06 \pm 2 \times 10^{-1}$ |
| pBrxXL$_{Sty}$-$\Delta pglX$ | $5.68 \times 10^{-1} \pm 3.94 \times 10^{-1}$ | $6.97 \times 10^{-1} \pm 2.28 \times 10^{-1}$ |
| pBrxXL$_{Sty}$-$\Delta pglZ$ | $2.75 \times 10^{-1} \pm 1.31 \times 10^{-1}$ | $5.60 \times 10^{-1} \pm 1.76 \times 10^{-1}$ |
| pBrxXL$_{Sty}$-$\Delta brxL$ | $7.50 \times 10^{-6} \pm 3.46 \times 10^{-6}$ | $9.43 \times 10^{-1} \pm 4.46 \times 10^{-1}$ |
| pBrxXL$_{Sty}$-$\Delta ariA$ | $3.65 \times 10^{-3} \pm 2.59 \times 10^{-3}$ | $9.07 \times 10^{-1} \pm 2.46 \times 10^{-1}$ |
| pBrxXL$_{Sty}$-$\Delta ariB$ | $8.15 \times 10^{-3} \pm 3.01 \times 10^{-3}$ | $8.67 \times 10^{-1} \pm 2.87 \times 10^{-2}$ |
| pBrxXL$_{Sty}$-$\Delta ariA\Delta ariB$ | $1.06 \times 10^{-2} \pm 2.96 \times 10^{-3}$ | $7.63 \times 10^{-1} \pm 4.99 \times 10^{-2}$ |

EOP ≥ $10^{-1}$
$10^{-1}$ > EOP ≥ $10^{-2}$
$10^{-2}$ > EOP ≥ $10^{-3}$
EOP < $10^{-3}$

**b**

| Genome Sample | Nanopore Average methylation (%) | PacBio Average methylation (%) |
|---|---|---|
| WGA[1] | 12.87 | 0 |
| p507 | 41.83 | 0 |
| pBrxXL$_{Sty}$ | 78.78 | 97.8 |
| pBrxXL$_{Sty}$-$\Delta brxA$ | 39.86 | 0 |
| pBrxXL$_{Sty}$-$\Delta brxB$ | 44.39 | 0 |
| pBrxXL$_{Sty}$-$\Delta brxC$ | 41.01 | 0 |
| pBrxXL$_{Sty}$-$\Delta pglX$ | 47.15 | 0 |
| pBrxXL$_{Sty}$-$\Delta pglZ$ | 49.09 | 0 |
| pBrxXL$_{Sty}$-$\Delta brxL$ | 87.54 | 97.7 |
| pBrxXL$_{Sty}$-$\Delta ariA$ | 68.45 | 99.8 |
| pBrxXL$_{Sty}$-$\Delta ariB$ | 87.30 | 99.8 |
| pBrxXL$_{Sty}$-$\Delta ariA\Delta ariB$ | 86.87 | 97.8 |

0
0 < % ≤ 30
30 < % ≤ 50
50 < % ≤ 70
70 < % ≤ 90
% > 90

[1]WGA, whole genome amplification

**Fig. 3 | Gene deletions within *Salmonella* BREX impact both phage defence and methylation. a** EOPs of TB34 and T7 tested against *E. coli* DH5α pBrxXL$_{Sty}$ WT and mutants, with *E. coli* DH5α pTRB507 as control. Values are mean EOPs from triplicate data, shown with standard deviation. **b** Detection of GATC**A**G N6mA motifs from genomic DNA of *E. coli* DH5α pBrxXL$_{Sty}$ WT and mutants, with *E. coli* DH5α pTRB507 (and whole genome amplified samples thereof) as controls. Data obtained using MinION and Pacific Biosciences (PacBio) sequencing.

PacBio sequencing (Fig. 3b)[43]. The PacBio results were more robust than those from MinION, with 0% of motifs modified in the WGA sample and 100% of motifs modified with pBrxXL$_{Sty}$ WT. The BREX mutants also showed either no, or near-saturated, methylation (Fig. 3b). The PARIS deletions resulted in close to WT levels of methylation by PacBio (Fig. 3b), indicating that PARIS is not involved in the observed methylation. These data show the genetic requirements for D23580 BREX-dependent host methylation and demonstrate the utility of two sequencing platforms when examining N6mA modifications.

### Structure of PglX shows SAM binding for methyltransferase activity

It has not been understood how BREX systems recognise their cognate motifs. The likely candidate protein, shown to be essential for methylation and defence, was the conserved PglX putative methyltransferase. In order to learn more about BREX motif recognition, the structure of *Salmonella* PglX was sought through X-ray crystallography. Following crystallisation and data collection, an Alphafold model of PglX was used as a search model for molecular replacement, assisting the solution and refinement of the crystallographic structure of *Salmonella* PglX bound to S-adenosyl-L-methionine (SAM), a co-

factor for methylation. The structure was refined using all data collected up to 3.5 Å (Fig. 4; Table 1).

The crystal structure contains two copies of PglX in the asymmetric unit, the smallest repeating unit of the crystal. However, the arrangement of the two copies allows only weak interactions that are likely formed due to interactions within the crystal rather than being biologically significant. The architecture of PglX presents two distinct domains, N-terminal and C-terminal, linked by a central short hinge region (residues 654–659) (Fig. 4a, b). Due to absence of available density, two short loop regions were unable to be modelled (residues 53–56 and 418–420), but otherwise the full PglX protein was resolved. SAM was also resolved bound within PglX (Fig. 4c).

The closest structural homologue for the solved PglX structure, as designated by the DALI server[44], remains the Type IIL restriction-modification system, MmeI[45] (PDB 5HR4; Z-score 20.3), though the N-terminal nuclease domain that is found in MmeI (and not PglX) is missing from the MmeI structure. As a result, whilst MmeI demonstrates both N6mA DNA methyltransferase and DNA restriction activity[45] the MmeI structure only has 60.8% sequence coverage against PglX, (1225 residues and 745 residues for PglX and MmeI, respectively), and aligns to PglX with an RMSD of 7.13 Å (2524 atoms; Supplementary Fig. 6a). The majority of this alignment falls

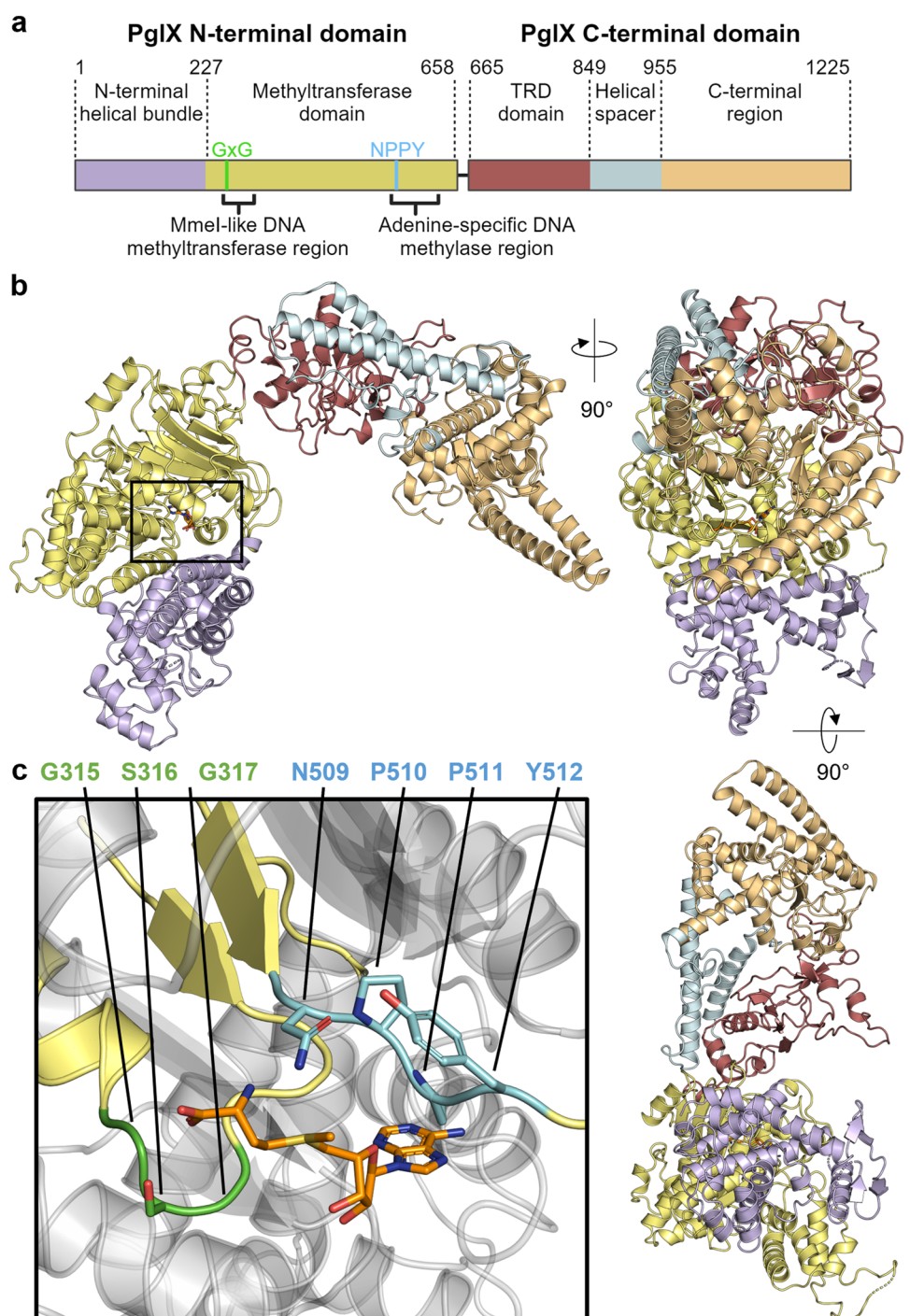

**Fig. 4 | PglX crystal structure shows methyltransferase and target recognition domains, with bound co-factor SAM. a** Schematic of PglX domain organisation with predicted MmeI-like DNA methyltransferase and adenine-specific DNA methylase regions denoted by square brackets. The methyltransferase domain (yellow) contains GxG (bright green) and NPPY (cyan) amino acid motifs responsible for S-adenosyl-L-methionine (SAM) binding and transfer of methyl group to an adenine residue, respectively, indicative of a γ-class amino-methyltransferase. The C-terminal region (light orange) is separated from the target recognition domain (TRD, red), by a long double helical spacer (light blue). The N-terminal domain contains an additional helical bundle (light purple) upstream of the methyltransferase domain. **b** Orthogonal views of PglX, shown as cartoon and coloured as per schematic in (**a**). **c** Close-up of the SAM binding region shown by box in (**b**). The SAM molecule sits between GSG and NPPY γ-class amino-methyltransferase amino acid motifs and is shown as orange sticks. Sidechains of the GSG and NPPY motifs are shown as sticks.

within the N-terminal domain of PglX and bridges the hinge region, extending into the C-terminal domain. The MmeI structure shows a methyltransferase domain bound to the SAM analogue sinefungin[45], and in our PglX structure SAM binds within the same pocket (Fig. 4). Within this homologous domain of PglX (residues 227–661) sit the amino-methyltransferase motif I GxG residues implicated in SAM

binding (residues 315–317), and adenine specific motif IV responsible for interacting with a flipped-out adenine base from the target DNA (NPPY; residues 509–512) (Fig. 4; Supplementary Fig. 6b). The presence and organisation of these motifs around the SAM molecule (Fig. 4c) is indicative of a γ-class amino-methyltransferase[46], consistent with its homology to MmeI[45]. Though MmeI has both

**Table 1 | X-Ray data collection and refinement statistics**

| Structure | PglX-SAM | PglX-SAM:Ocr |
|---|---|---|
| PDB Code | 8C45 | 8Q56 |
| Wavelength | 0.9795 | 0.9795 |
| Resolution range | 70.65–3.50 (3.61–3.5) | 59.61–3.50 (3.63–3.5) |
| Space group | P 41 21 2 | C 1 2 1 |
| Unit cell, *a b c* (Å), | | |
| α β γ (°) | 138.54 138.54 407.96 90 90 90 | 238.46 60.79 146.64 90 114.89 90 |
| Total reflections | 95636 | 47094 |
| Unique reflections | 51157 (4333) | 24556 (2426) |
| Multiplicity | 1.9 | 1.9 |
| Completeness (%) | 86.8 | 95.8 |
| Mean I/sigma(I) | 6.9 (0.2) | 3.8 (0.3) |
| $R_{merge}$ | 0.05 | 0.028 |
| $R_{meas}$ | 0.06 | 0.09 |
| $CC_{1/2}$ | 0.99 (0.32) | 0.99 (0.38) |
| Reflections used in refinement | 45528 (912) | 24038 (1957) |
| Reflections used for $R_{free}$ | 2344 (50) | 1922 (144) |
| $R_{work}$ | 0.265 (0.444) | 0.250 (0.407) |
| $R_{free}$ | 0.278 (0.480) | 0.292 (0.420) |
| Number of non-hydrogen atoms | 19892 | 10776 |
| macromolecules | 19848 | 10747 |
| ligands | 98 | 49 |
| solvents | 0 | 2 |
| Protein residues | 2432 | 1318 |
| RMS (bonds, Å) | 0.005 | 0.004 |
| RMS (angles, °) | 0.91 | 0.78 |
| Ramachandran favoured (%) | 90.36 | 91.6 |
| Ramachandran allowed (%) | 9.64 | 8.4 |
| Ramachandran outliers (%) | 0 | 0 |
| Average B-factor | 167.0 | 138.0 |
| macromolecules | 168.0 | 138.5 |
| ligands | 114.0 | 139.0 |
| solvent | N/A | 113.43 |

Values in parenthesis are for the highest-resolution shell.

methyltransferase and restriction activities the MmeI nuclease domain (residues 1–155) was not resolved in the MmeI structure[45]. The nuclease domain of MmeI is separated by a helical linker. The N-terminal domain of PglX contains a similar linker and an N-terminal helical bundle (residues 1–227), but no nuclease domain (Fig. 4a, b). Assessing conservation between homologues in the UniRef database using ConSurf[47], the MmeI-like DNA methyltransferase region of PglX appears highly conserved compared to the N-terminal helical bundle domain (Supplementary Fig. 6c). Using DALI to search for structural homologues of the C-terminal domain alone (residues 672–1221) returns Type I RM specificity subunits. The immediate section of the C-terminal domain of PglX aligns with target recognition domains (TRD) required for motif binding (residues 662–849). This is followed by two long spacer helices (residues 850–960) that mimic dimerised spacers found in specificity factors of Type I DNA methyltransferases such as EcoKI[48] (Fig. 4a, b). The spacers lead to a final C-terminal region of unknown function (residues 961–1225). Interestingly, the spacer and C-terminal regions extend 320 residues beyond the end of the

alignment with MmeI and show a high degree of conservation (Fig. 4a, b; Supplementary Fig. 6a, c). This might suggest a specialised function conserved to allow BREX activity, perhaps as a binding surface for other BREX components. As a result, the PglX structure, and lack of nuclease motifs and potential aligned catalytic residues, supports PglX acting as a methyltransferase only, and not acting as a restriction enzyme.

With expression and purification methods established, and the structure supporting PglX as the BREX methyltransferase (Fig. 4), a SAM-dependent methyltransferase assay was performed to assess the ability of purified PglX to methylate DNA in vitro. Using *E. coli* DH5α genomic DNA known to contain the target BREX$_{Sty}$ motif as a substrate, PglX was added and incubated for 30 min at room temperature in a buffer containing SAM. Methyltransferase activity was measured indirectly via the reaction product, S-adenosyl-L-homocysteine (SAH). No methylation was apparent from PglX under these conditions (Supplementary Fig. 8), nor when we added in purified PglZ and BrxB. We hypothesise that PglX methyltransferase activity likely requires the presence of other BREX components, but the combination and ratio remains to be optimised.

### *Salmonella* BREX can be inhibited by Ocr homologues through binding PglX

Ocr is the T7-encoded restriction system inhibitor that blocks phage defence activity of the *E. coli* BREX system[27]. It has recently been shown that Ocr competes with DNA for PglX binding[49]. Additionally, Ocr triggers Abi by the type II PARIS phage defence system[16]. BREX$_{Sty}$ also encodes a homologue of PARIS (Fig. 1a). Though notably, no activity was observed for BREX$_{Sty}$ against phage T7 (Fig. 2 and Fig. 3a). Following the production of individual gene knockouts, it was possible to individually assay inhibition of BREX and activation of PARIS by Ocr. To determine whether Ocr inhibited BREX, vector pBAD30-*ocr* was generated. EOP assays were then carried out with *E. coli* DH5α pBrxXL$_{Sty}$-ΔariAΔariB pBAD30-*ocr* and showed that expression of Ocr fully inhibited BREX defence (Fig. 5a). As Ocr is a product of T7, a coliphage, this experiment was also repeated using an Ocr homologue, Gp5, encoded by *Salmonella* phage Sp6[50]. Homology was inferred by protein sequence searches using BLAST (NP_853565.1: 78.6% sequence similarity, 88% coverage) followed by predictive modelling from protein sequence using AlphaFold[51]. The structures of Ocr and Gp5 aligned with an RMSD of 0.91 Å (652 atoms). We again selected TB34 as a model phage and tested Gp5 activity. Results showed that Gp5 also fully inhibited the phage defence mediated by pBrxXL$_{Sty}$ (Fig. 5a).

As we had demonstrated inhibition of BREX by overexpression of the inhibitors Ocr and Gp5, it was postulated that the same experimental system might elicit phage defence mediated by the PARIS system. This time, the pBrxXL$_{Sty}$-Δ*pglX* strain was used for co-expression of Ocr or Gp5, as this strain is deficient for BREX phage defence but retains the PARIS system. The resulting EOP assays did not show PARIS-dependent defence activity against TB34 (Supplementary Fig. 8). We are therefore yet to find conditions that stimulate activity of the *Salmonella* PARIS system.

We then aimed to recreate a PglX:Ocr complex[27] using our purified *Salmonella* PglX and Ocr, and visualise the resulting complexes with analytical SEC. Elution volume is dependent on protein molecular weight, and can also reflect the shape and size of the protein molecule itself. The hydrodynamic radius of protein samples seen by analytical SEC can be calculated from the observed K$_{av}$ value[52], allowing comparison to the calculated hydrodynamic radius of predicted protein and protein complex models produced by AlphaFold[53]. The solution state of native PglX was determined using analytical SEC. PglX eluted from the SEC column at 11.2 ml (Supplementary Fig. 9a), which indicated a hydrodynamic radius of 54.8 Å, matching the 57 Å calculated hydrodynamic radius of PglX. These data indicate that PglX exists as a

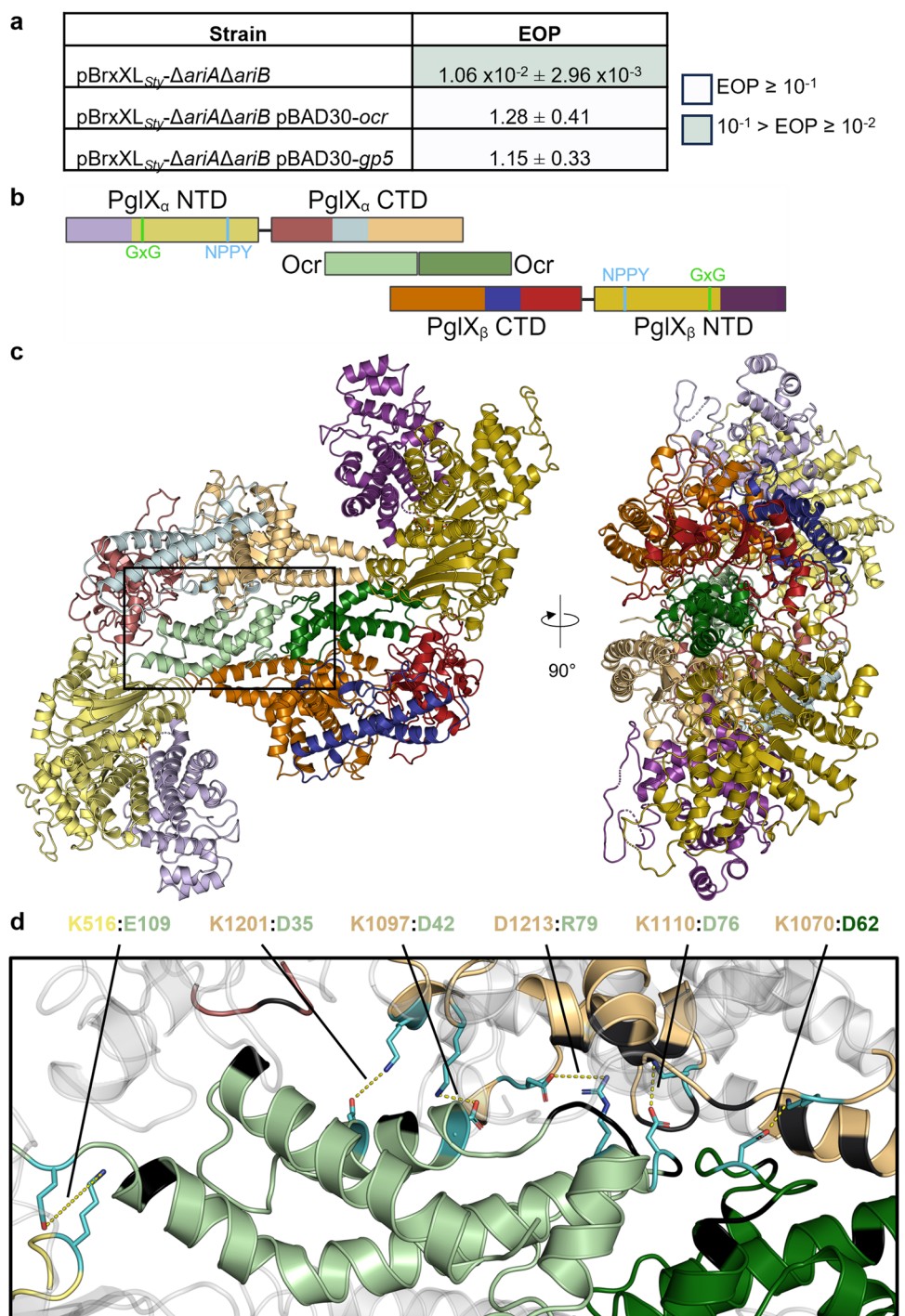

| Strain | EOP |
|---|---|
| pBrxXL$_{Sty}$-ΔariAΔariB | $1.06 \times 10^{-2} \pm 2.96 \times 10^{-3}$ |
| pBrxXL$_{Sty}$-ΔariAΔariB pBAD30-*ocr* | $1.28 \pm 0.41$ |
| pBrxXL$_{Sty}$-ΔariAΔariB pBAD30-*gp5* | $1.15 \pm 0.33$ |

EOP ≥ $10^{-1}$
$10^{-1}$ > EOP ≥ $10^{-2}$

**Fig. 5 | Ocr inhibits BREX defence by forming a heterotetrameric complex with PglX. a** EOPs of TB34 against *E. coli* DH5α strains carrying pBrxXL$_{Sty}$-ΔariAΔariB and induced plasmids for Ocr and Gp5, with *E. coli* DH5α pTRB507 as control. Values are mean EOPs from triplicate data, shown with standard deviation. **b** Schematic showing PglX domains relative to bound Ocr dimer in a heterotetrameric complex. The two Ocr protomers in the dimer are shown in pale green and dark green. Domain colourings of PglX are as described in Fig. 4, with darker shades of respective colours used for each domain within the second PglX molecule in the complex. **c** Orthogonal views of the PglX-SAM:Ocr complex structure, shown as cartoons and coloured as per (**b**). **d** Interactions between PglX and Ocr, close-up of the boxed region in (**c**). Residues involved in the formation of salt bridges between PglX and Ocr molecules are labelled and shown in cyan, with sidechains. Residues forming hydrogen bond interactions are shown in black.

monomer in solution, supporting our conclusions from the PglX-SAM structure (Fig. 4). The Ocr sample was then examined by analytical SEC in isolation (Supplementary Fig. 9a). The Ocr SEC profile gave a single species at 14.6 ml, with a calculated hydrodynamic radius of 30.4 Å. Ocr is known to be a dimer in solution[26,54], which would be 27.6 kDa and corresponds to a calculated hydrodynamic radius of 24.3 Å. Purity of the Ocr sample was confirmed by mass spectrometry and SDS-PAGE (Supplementary Figs. 9b and c). PglX and Ocr were then combined at a 1:2 molar ratio prior to SEC (Supplementary Figs. 9a and b). The combined sample produced an additional peak at 10.3 ml beyond those from the individual PglX and Ocr samples (Supplementary Figs. 9a and b) and moved the bulk of the PglX peak. The peak at 10.3

ml indicated a large complex of approximately ~391 kDa, potentially comprised of at least two copies of PglX, and Ocr dimers (Supplementary Fig. 9a). A model of two monomers of PglX and one Ocr dimer produced by AlphaFold produced a predicted hydrodynamic radius of 58.3 Å, compared to a calculated hydrodynamic radius of 63.9 Å for the observed A-SEC peak. This suggested that the additional peak eluting at 10.3 ml represented a PglX-Ocr heterotetramer in solution. The protein contents of each peak were examined by SDS-PAGE, and the results matched our assignments, wherein PglX and Ocr could be found to have shifted and co-elute at 10.3 ml (Supplementary Fig. 9b).

### PglX forms a heterotetrameric complex with inhibitor Ocr

To investigate the mechanism of BREX inhibition by Ocr, efforts were made to produce a structural model via X-ray crystallography. PglX-SAM and Ocr were mixed at a 1:2 molar ratio and incubated prior to setting crystallisation trials. After data collection and merging, and using our previously derived PglX-SAM structure (Fig. 4) and the PDB structure of Ocr (1S7Z) as search models, the PglX-SAM:Ocr structure was solved to 3.5 Å (Fig. 5b, c; and Table 1).

Within the asymmetric unit, PglX-SAM binds to a protomer of Ocr as a 1:1 complex, with the single protomer of Ocr binding along the positively charged region of the C-terminal domain of PglX. Data on the solution state of Ocr (a dimer), coupled with our predictions of complex size by analytical SEC, indicated that PglX:Ocr should form a larger complex. Indeed, when we searched for crystallographic symmetry mates that showed packing of PglX-SAM:Ocr, the predicted complex was visible (Fig. 5b, c). In this complex, the Ocr protomers perfectly align and abut one another, forming the equivalent of a solution state dimer, and the size matches our analytical SEC. We therefore concluded that this heterotetrameric form represented the solution state of the PglX-SAM:Ocr complex (Fig. 5b, c).

Within PglX, there were again two regions of the sequence which could not be modelled due to insufficient density (residues 54–55 and 413–420). The latter is an extended gap in the same region as a smaller gap in the PglX-SAM structure (D418 – F420), suggesting flexibility in this region. Also visible in the PglX-SAM:Ocr structure is a bound SAM molecule, in the same ligand binding position as seen in the PglX-SAM structure (Figs. 4 and 5). Though all three modelled SAM ligands across the two structures fit into the same pocket within PglX, the exact orientation of ribose and methionine components of the SAM ligands varied, likely due to the resolution of our models (Supplementary Fig. 10). The PglX molecules from the PglX-SAM and PglX-SAM:Ocr structures align closely with an RMSD of 1.34 Å (9564 atoms), suggesting that binding of Ocr does not elicit any substantive domain movement (Supplementary Fig. 11). EMBL PISA[55] analysis of the complex showed that there was no dimerisation of PglX in the complex, and identified important residues for Ocr interactions. The complex is stabilised by a number of hydrogen bonds between Ocr and the C-terminal domain of PglX (Fig. 5d). Six salt bridges are produced between R79, D35, D42, D62, D76 and E109 of Ocr and D1213, K1201, K1097, K1070, K1110, and K516 of PglX, respectively (Fig. 5d). Though no movement is observed in PglX, the binding of Ocr to Type I RM complexes elicits domain movement similar to DNA binding, suggesting either that PglX domain movement is reliant on interactions with other BREX components, or that DNA binding occurs along the C-terminal domain prior to movement towards the methyltransferase N-terminal domain. If other BREX components are required for such movement, the finding would be consistent with the lack of methyltransferase activity in vitro in the absence of other BREX components (Supplementary Fig. 7) or the lack of methyltransferase activity from PglX alone in vivo[25]. Collectively, these data suggest that Ocr acts as a DNA mimic, capable of sequestering PglX and therefore blocking BREX activity by preventing recognition of target DNA.

### Structural comparisons show multiple potential modes of DNA binding by PglX

Ocr mimics the structure of 20–24 bp of bent B-form DNA[26], as shown by the binding of both molecules to the EcoKI methyltransferase complex[48]. Using the DNA-bound (PDB 2Y7H) and Ocr-bound (PDB 2Y7C) complexes of EcoKI, the Ocr and DNA molecules were superimposed onto each other. As a result, the Ocr molecule in the PglX-SAM:Ocr structure was aligned with the Ocr molecule in 2Y7C, effectively aligning the B-form DNA from 2Y7H to the Ocr molecule in PglX-SAM:Ocr structure (Supplementary Fig. 12a). There does appear to be enough space for an extended DNA molecule to pass through the groove in the hinge region in this orientation, but Ocr is not long enough to extend through this region (Fig. 6a, b; and Supplementary

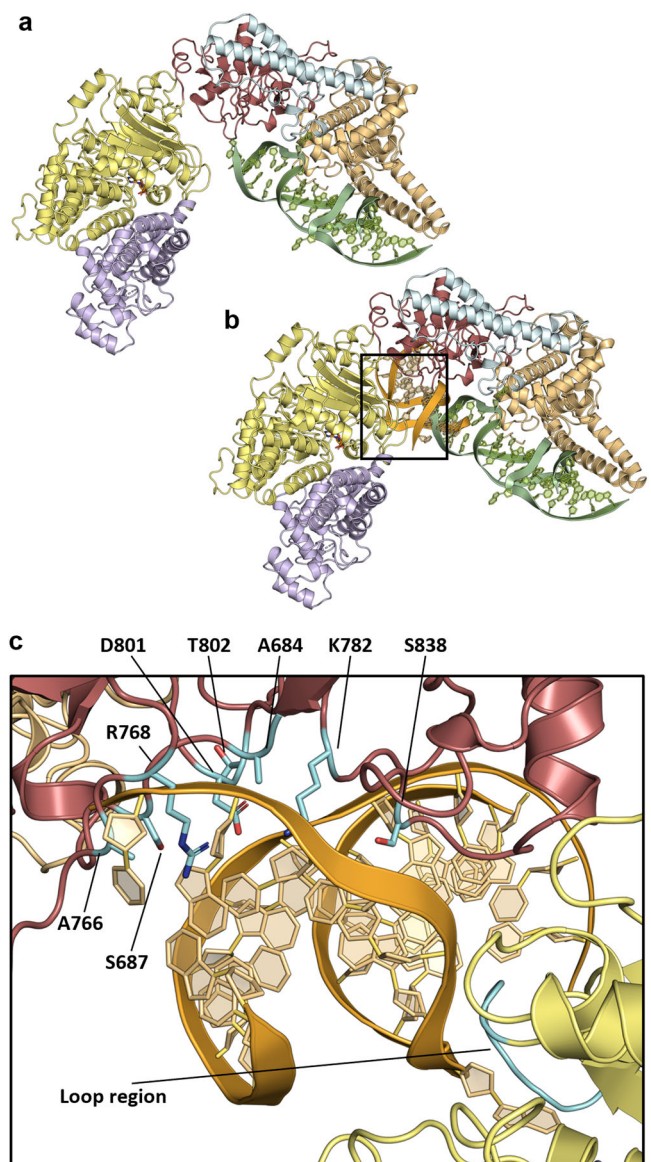

**Fig. 6 | Structural comparisons with Ocr-bound complexes and MmeI suggest differing PglX DNA binding modes. a** DNA molecule (green) representative of Ocr dimer (PDB codes 2Y7C and 2Y7H) superimposed onto the bound Ocr dimer within the PglX-SAM:Ocr complex. **b** The DNA molecule (orange) from MmeI (5HR4) superimposed onto PglX, showing a different potential binding position and angle to that implied by Ocr DNA (green) in (**a**). **c** Positions of residues aligning with those targeted for mutation when altering MmeI DNA motif specificity (cyan) relative to the superimposed MmeI DNA molecule (orange) from (**b**). PglX domains coloured as per Fig. 4.

Fig. 12b). This implicates the C-terminal domain in DNA binding, though raises the possibility of an alternative DNA binding orientation.

The surface charge of PglX was calculated using APBS software plugin[56] and modelled in PyMOL[57] to attempt to predict alternate DNA binding positions (Supplementary Fig. 13a). Notably, PglX displayed a large positively charged surface area in the hinge region between the N-and C-terminal domains, extending further along the inside of the C-terminal domain. As MmeI was solved in a DNA-bound state (PDB 5HR4), we could superpose these two structures and remove MmeI, leaving the DNA molecule sat within the positively charged hinge region of PglX (Fig. 6b; and Supplementary Fig. 13b). Notably, the angle of the superimposed DNA molecule from the MmeI structure (PDB 5HR4) differs from the previously identified angle of the 2Y7C DNA molecule (Fig. 6b). Further to this, the DNA molecule from the MmeI structure contained an adenine base which had been flipped out of the DNA molecule, in preparation for methyl transfer. Looking at the position of the superimposed MmeI DNA molecule, this adenine base is positioned close to the SAM molecule in PglX (Supplementary Fig. 13b). Together, these data suggest that PglX might bind DNA within the hinge region in a similar conformation to that seen with MmeI, though the exact orientation of the DNA molecule may shift around the position of the adenine base. In support of this prediction, the donated methyl group of the SAM is not quite positioned correctly for transfer to the flipped adenine (Supplementary Fig. 13b). In this model, unlike for Ocr mimicking DNA, the distal C-terminal region of PglX remains largely removed from the DNA molecule, though binding of DNA may require, or produce, a conformational change in PglX that brings this domain closer to the DNA.

## DNA and Ocr binding by PglX is mediated through charged residues on the CTD

We generated mutant versions of PglX based on our comparisons of DNA-bound complex predictions (Fig. 6) and our Ocr-bound complex (Fig. 5). Deletion of the entire C-terminal domain allowed us to express and purify just the PglX N-terminal domain (PglX NTD) (Fig. 4a). Though we did generate constructs to also express and purify the PglX C-terminal domain alone, it was not possible to express the CTD. We then made a multi-site mutant, targeting the CTD. As per (Fig. 5d), we selected residues identified as important for Ocr binding. The mutant construct included six mutations; K616A, K1201A, K1097A, D1213A, K1110A and K1070A. The resulting protein was denoted PglX[Mutant].

Mass photometry (Refeyn) was used to assess PglX WT, PglX NTD and PglX[Mutant] (Fig. 7a; Supplementary Fig. 14). All three variants were observed at approximately the expected weight for a monomer, though there was some evidence for dimer formation by PglX[Mutant] (Fig. 7a). Mass photometry was then used to examine DNA-binding, using a range of 120 bp dsDNA probes (Fig. 7a). DNA 1 contains two BREX motifs, each on opposing strands, therefore reading towards one another. DNA 2 contains two BREX motifs on the same strand, therefore reading in the same direction. DNA 3 contains a single BREX motif, and DNA 4 contains no BREX motifs. Using PglX WT we observed multiple DNA-bound species, with weights matching a 1:1 PglX WT:DNA complex, and a 2:1 PglX WT:DNA complex (Fig. 7a). These were clearly evident for all four DNA probes. A larger 3:1 PglX WT:DNA complex was observed for DNA 3 and DNA 4, suggesting that fewer or an absence of BREX motifs allowed non-specific binding to build up larger complexes (Fig. 7a). In contrast, no additional species were observed when using any DNA probe for the PglX NTD and PglX[Mutant] proteins (Fig. 7a). Due to the size of the PglX NTD, the observed peaks (Fig. 7a) may represent a mixture of PglX NTD and non-interacting DNA. Collectively, these data indicate that charged surface residues of the CTD are essential for DNA-binding by PglX.

The same PglX proteins were then tested for interactions with Ocr using mass photometry (Fig. 7b). PglX WT formed 1:2 and 2:2 PglX WT:Ocr complexes (Fig. 7b). PglX NTD and PglX[Mutant] were unable to form complexes with Ocr (Fig. 7b). This indicates the same regions used for DNA binding are also critical for Ocr binding, as predicted.

## PglX can be rationally engineered to alter phage target and methylation motif

Rational engineering of PglX could potentially allow for a BREX system to be targeted against a different set of phages, and for the generation of specific methylation patterns. To this end, protein sequences from BREX-related methyltransferases with assigned DNA recognition motifs were collected and added to the sequences of BREX methyltransferases identified in the REBASE RM database[58]. BLASTp was then used to find 32 distinct sequences that displayed high sequence similarity scores to PglX (<E100) (Supplementary Fig. 15). Most of the predicted motifs from REBASE were inferred by matching the BREX methyltransferase to an N6mA modification observed in genomic sequencing data. MmeI is the closest structural homologue of PglX and the residues essential for motif recognition have been identified from structural data[45]. As with PglX, MmeI recognises a 6 bp motif (TCCR**A**C) and produces N6mA modifications at the 5th adenine base. Structural alignments of MmeI and PglX allowed identification of the residues of PglX that aligned with the residues involved in MmeI motif recognition and suggested regions in which to focus the search for covariation in BREX methyltransferase sequence alignments. Candidate residues and alterations were then chosen based on these alignments. For example, for motif position -1 (relative to the modified adenine base); lysine was conserved at residue 802 for enzymes recognising cytosine at this position, or histidine was conserved at residue 838 for enzymes recognising guanosine at this position, or asparagine was conserved at residue 838 for enzymes recognising adenine at this position (Supplementary Fig. 15). We designed 23 mutants that altered all five of the non-modified base positions in the PglX recognition motif (Supplementary Table 1). The regions targeted for mutation were overlaid on our structures and shown to gather mainly within the TRD of PglX (between residues 684–838), with one additional loop (residues 591–600) within the methyltransferase domain (Fig. 6c).

Following the design of the PglX mutants, an assay system was required to test function. Generating each of the mutants individually in the 17.9 kb pBrxXL*Sty* plasmid would have been costly and time consuming. Instead, a complementation system was designed that utilised the pBrxXL*Sty*-Δ*pglX* construct. The BREX*_Sty_* *pglX* gene was cloned into pBAD30. Complementation of the pBrxXL*Sty*-Δ*pglX* construct with the pBAD30-*pglX* plasmid in EOP assays provided phage defence against TB34, albeit slightly lower than that seen from the *E. coli* DH5α pBrxXL*Sty* construct (Fig. 8a). Next, a marker was required to indicate whether the recognition motif had been modified. Again, it was preferable to initially test this through functional EOP assays as sequencing for methylation changes caused by all 23 mutants would be laborious and expensive. Fortunately, the activity of pBrxXL*Sty* had already been characterised against the Durham Phage Collection and phages in this collection had been sequenced to allow enumeration of BREX recognition motifs[33]. This allowed the identification of one phage, Trib, which was susceptible to *E. coli* and *E. fergusonii* BREX systems but contained no native *Salmonella* D23580 BREX recognition motifs and therefore was not impacted by BREX*_Sty_* (Fig. 8a)[33]. Trib did, however, encode all of the predicted re-engineered motifs (Supplementary Table 1). This finding allowed us to first screen all mutants for phage defence activity against phage Trib before determination of the recognition motif of any active mutants by sequencing.

EOP assays were carried out in triplicate for all 23 pBAD30-*pglX* mutants co-expressed with the pBrxXL*Sty*-Δ*pglX* construct in *E. coli* DH5α (data not shown). Mutant 3 appeared to provide around 10-fold protection against Trib (Fig. 8a), similar to phage defence levels provided by BREX*_Eferg_* against this phage[33]. Mutants 8, 10, 15 and 22 showed sporadic reductions in EOP, usually around two-fold. Mutant 4

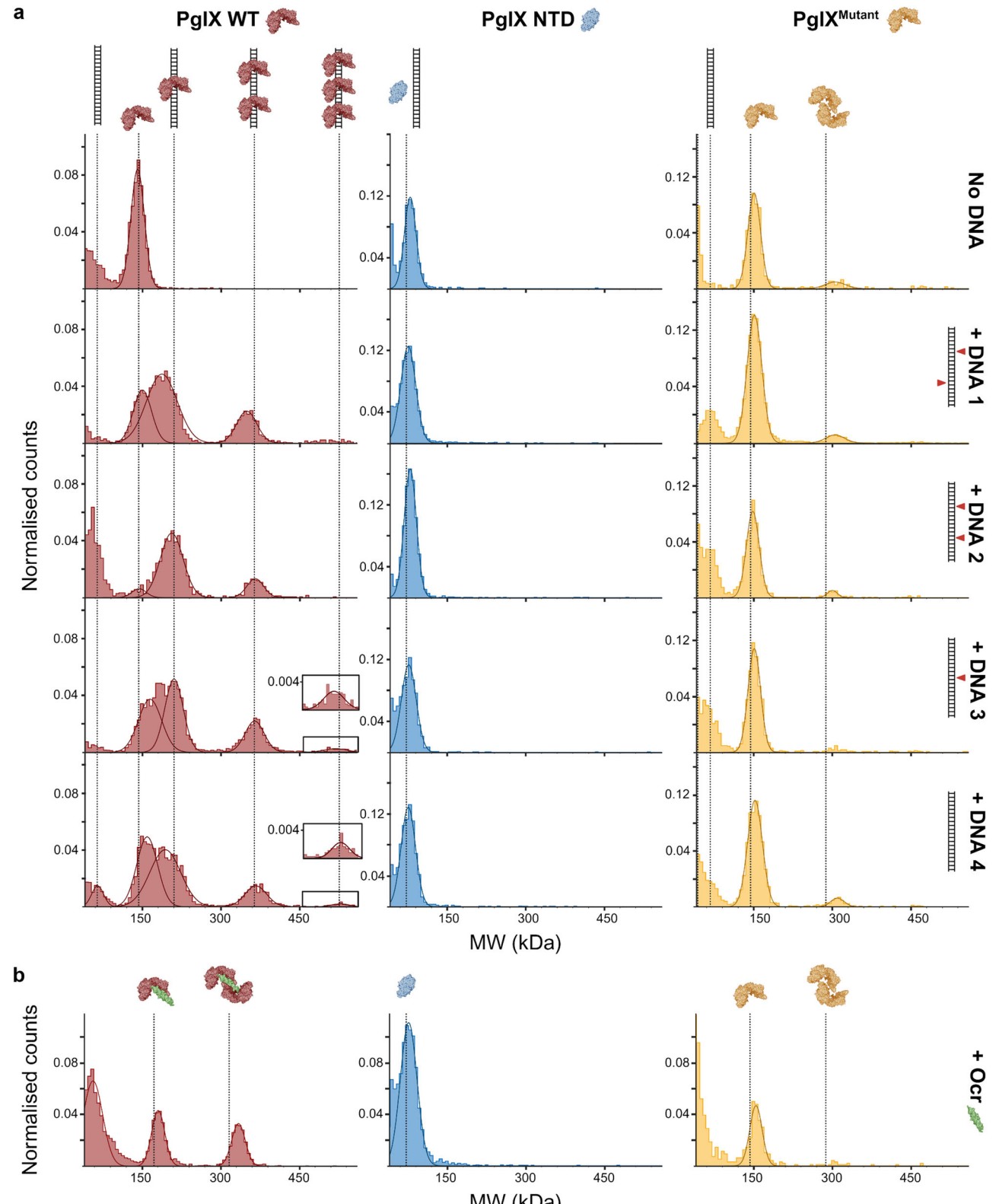

**Fig. 7 | PglX binds DNA and Ocr through charged residues on the CTD. a** Mass photometry (Refeyn) assays examining the binding of PglX WT (red), PglX NTD only (blue) and PglX^Mutant (yellow), in the absence or presence of 120 bp dsDNA substrates. DNA 1, two BREX motifs, one on each strand; DNA 2, two BREX motifs, both on same strand; DNA 3, on BREX motif; DNA 4, no BREX motifs. Insets show larger complexes formed with DNA 3 and DNA 4 for PglX WT. **b** Mass photometry of proteins in (**a**) binding to Ocr (green). Positions of resulting complexes are indicated by the presented cartoons and dashed vertical lines. Data are representative of three independent experiments. Data on raw counts and gaussian standard deviation are available in Supplementary Fig. 14.

**a**

| | | | TB34 | Trib |
|---|---|---|---|---|
| **# Genomic Motifs** | | Wild type (GATC**A**G) | 120 | 0 |
| | | *pglX* mut.3 (GAT**A**A**G**) | 93 | 83 |
| **WT EOP** | | DH5α pBrxXL*Sty* | $2.86 \times 10^{-2} \pm 1.54 \times 10^{-2}$ | $1.19 \pm 0.11$ |
| | | DH5α pBrxXL*Sty*-Δ*pglX* + pBAD30-*pglX* | $9.63 \times 10^{-2} \pm 6.14 \times 10^{-2}$ | $7.52 \times 10^{-1} \pm 0.035$ |
| **Mut.3 EOP** | | DH5α pBrxXL*Sty*-Δ*pglX* + pBAD30-*pglX* (mut.3) | $1.45 \times 10^{-1} \pm 1.33 \times 10^{-1}$ | $9.98 \times 10^{-2} \pm 1.02 \times 10^{-1}$ |
| | | DH5α pBrxXL*Sty* (*pglX* mut.3) | $1.03 \times 10^{-1} \pm 9.13 \times 10^{-2}$ | $4.28 \times 10^{-2} \pm 4.33 \times 10^{-2}$ |

Legend:
- EOP $\geq 5 \times 10^{-1}$
- $5 \times 10^{-1} >$ EOP $\geq 10^{-1}$
- $10^{-1} >$ EOP $\geq 5 \times 10^{-2}$
- EOP $< 5 \times 10^{-2}$

**b**

| Construct | Motif | Motif Sites in DH5α genome | Percentage Methylated (%) |
|---|---|---|---|
| DH5α pBrxXL*Sty* | GAT**C**AG | 2947 | 97.8 |
| | GAT**A**AG | 2346 | 0 |
| DH5α pBrxXL*Sty*-Δ*pglX* + pBAD30-*pglX* | GAT**C**AG | 2947 | 99.4 |
| | GAT**A**AG | 2346 | 0 |
| DH5α pBrxXL*Sty*-Δ*pglX* + pBAD30-*pglX*(mut.3) | GAT**M**AG | 5293 | 99.6 |
| DH5α pBrxXL*Sty* (*pglX* mut.3) | GAT**M**AG | 5293 | 99.8 |

**Fig. 8 | PglX is the sole specificity factor for BREX and can be rationally engineered to re-target BREX methylation and phage defence. a** EOP results of phages TB34 and Trib tested against *E. coli* DH5α pBrxXL*Sty* WT and PglX mut.3 in the context of BREX*Sty*, or against DH5α pBrxXL*Sty*-Δ*pglX* with complementation plasmid pBAD30-*pglX* (WT) or pBAD30-*pglX* (mut.3). Values are mean EOPs from triplicate data, shown with standard deviation. **b** PacBio sequencing results showing genomic methylation in strains as described in (**a**).

consistently produced poor overnight growth and failed to provide sufficient bacterial lawns for plaque enumeration, even after increasing the inoculum volume. Remaining mutants demonstrated no noticeable reduction in plaquing efficiency. To confirm the BREX system remained functional against other targets, mutants 3, 8, 10, 15 and 22 were also assayed against phage TB34. Mutant 3 caused a reduction in EOP for TB34 similar to that shown against Trib, though around twofold higher than produced by the *E. coli* DH5α pBrxXL*Sty* strain (Fig. 8a). The remaining 18 mutants did not show any reduction in EOP against TB34, despite TB34 encoding the expected re-engineered motifs, and were deemed to be inactive. There was also a small reduction in BREX activity in the complemented system (Fig. 8a). Accordingly, the T802A and S838N mutations of mutant 3 were also generated directly within the *pglX* gene of pBrxXL*Sty*, resulting in pBrxXL*Sty*(*pglX* mut.3) that did not require complementation. This new construct was assayed against both TB34 and Trib. Now in context within the BREX locus, EOP values were reduced further for both TB34 and Trib against *E. coli* DH5α pBrxXL*Sty*(*pglX* mut.3), though still not quite as low as shown by the activity of the WT BREX system against TB34 (Fig. 8a).

Next, the host genomes of *E. coli* DH5α pBrxXL*Sty*(*pglX* mut.3) and *E. coli* DH5α pBrxXL*Sty*-Δ*pglX* + pBAD30-*pglX*(mut.3) strains were sequenced and genomic methylation levels were assessed by PacBio sequencing, alongside the WT strains (Fig. 8b). The *E. coli* DH5α pBrxXL*Sty*-Δ*pglX* + pBAD30-*pglX* control had almost 100% methylation at GATC**A**G sites, and no methylation at GATA**A**G sites, demonstrating that the complementation system mediated efficient methylation of WT motifs, as observed for pBrxXL*Sty* (Fig. 8b). Analysis of the mutant 3 strains revealed methylation at almost 100% of GAT**M**AG motifs, generated by the PacBio software as a combined output for detection of two methylated motifs (Fig. 8b). This indicated that the mutations of mutant 3, T802A and S838N, had not altered the recognised motif to

GATA**A**G as predicted, but had broadened recognition to include both the original GATC**A**G motif and also GATA**A**G, resulting in this new motif GAT**M**AG. These data collectively demonstrate the successful re-engineering of PglX to target BREX against new phages, and to methylate altered DNA sequence motifs. The experiments also demonstrated that PglX is the specificity factor in the BREX phage defence system, providing motif recognition for both phage targeting and host methylation.

## Discussion

This study provides microbiological, genetic and epigenomic characterisation of the BREX phage defence island within *Salmonella* D23580. We present the first structures of the putative PglX methyltransferase, bound to SAM and in complex with the phage-derived inhibitor Ocr. Finally, though not a trivial task based on the number of mutants tested, we demonstrate successful rational engineering of BREX, opening up the potential for tailored phage targeting and generation of specific N6mA motifs. This work identifies PglX as the specificity factor for methylation and phage defence within BREX.

Clustered phage defence systems can provide additive[22] or even synergistic[59] protection. The *Salmonella* D23580 BREX phage defence island has an embedded PARIS system (Fig. 1a), suggesting a complementary relationship; PARIS has been shown to cause abortive infection upon encountering the phage encoded anti-restriction protein, Ocr, which in turn inhibits BREX defence in *E. coli*[16,27]. Using an *E. coli* model, we saw no activity from the *Salmonella* BREX phage defence island against Ocr-encoding phage T7 (Fig. 2). The reason that BREX*Sty* had no impact was because T7 does not encode any GATC**A**G motifs. PARIS also did not respond to Ocr (Fig. 2). Using an Ocr homologue from a *Salmonella* phage also did not activate PARIS (Supplementary Fig. 8), and so we can only conclude that the PARIS

system may provide protection, but that a susceptible phage has not yet been tested.

As with previous studies, *Salmonella brxB, brxC, pglX* and *pglZ* proved essential for both restriction and methylation (Fig. 3)[17,25]. However, *brxA* was required for phage defence and methylation in *Salmonella* BREX (Fig. 3) and *Acinetobacter* BREX[17], but was shown to be dispensable for both activities in *E. coli* BREX[25]. BrxA is a DNA-binding protein[23] with an unknown role in BREX activity, so we are yet to understand the variable requirement for *brxA. Salmonella brxL* was demonstrated to be dispensable for host methylation (Fig. 3b) and this matches the observed phenotype in *Acinetobacter* and *E. coli*[17,25]. Curiously, whilst *brxL* was essential for phage defence in both *E. coli* and *Acinetobacter* BREX systems[17,25], it was not required for *Salmonella* BREX (Fig. 3a). BrxL was recently shown to form a dimer of hexameric rings, forming a barrel-like structure that binds and translocates along DNA[24]. Thus, BrxL had been considered to have an essential role as the "effector" for BREX phage defence. Clearly this is not the case in the *Salmonella* BREX system, as made more apparent by EOP results for *E. coli* DH5α pBrxXL*Sty*-Δ*brxL* tested against the Durham phage collection (Supplementary Fig. 4a)[33], and when testing a *brxL* deletion in the original host *Salmonella* strain (Supplementary Fig. 4c). Deletion of *brxL* even enhanced protection for certain phages (Supplementary Figs. 4a and c). It is possible that *Salmonella* BrxL modulates or regulates BREX activity in some way. RM systems are often associated with restriction alleviation proteins that activate in times of stress, reducing restriction activity and increasing methylation activity; a phenotype characteristic of Type I RM systems[60–62]. It is possible that BrxL plays an analogous role to restriction alleviation proteins within BREX and that defence activity increases in the absence of BrxL. However, if that were the case, why is this phenotype not observed for *brxL* deletions in *E. coli* or *Acinetobacter* BREX systems? Overexpression of a C-terminal fragment of BrxL has been shown to upregulate several genes elsewhere in the *Salmonella* genome, including certain prophage genes[29]. It was postulated that because the corresponding Lon-like domain in the C-terminal BrxL fragment has similarity to the Lon-related C-terminal domain of RadA that is required for DNA branch migration in homologous recombination[63], BrxL may inhibit phage DNA replication at DNA forks. This would be somewhat in keeping with the model of BrxL complexes translocating along DNA. The *brxL* deletion data provide additional insight to this model as they suggest that whilst BrxL-dependent BREX defence may interrupt replication forks, other BREX components have another activity sufficient to prevent phage DNA replication.

To better understand the activity of other BREX components we produced the first structure of PglX, demonstrating that the N-terminal domain has a methyltransferase fold, and binds SAM (Fig. 4). In contrast, fold, conserved residues, and surface properties of the C-terminal domain suggest a role in DNA recognition and binding. Despite repeated efforts we could not crystallise PglX with DNA. We hypothesised that Ocr binding might provide insight into DNA binding by PglX. We showed that Ocr and *Salmonella* homologue Gp5 both impacted BREX phage defence (Fig. 5a), and produced stable complexes of PglX:Ocr (Supplementary Fig. 9a). The resulting structure involved the interaction of an Ocr dimer with two PglX monomers (Fig. 5b, c). The structure of PglX in the Ocr-bound complex varied little in comparison to the PglX-SAM structure, and there was no movement of domains upon Ocr binding. Using these two structures, we developed two models for DNA binding by PglX, via (i) alignment with a 20 bp DNA molecule represented by Ocr and (ii) alignment via DNA bound to MmeI (Fig. 6a, b; and Supplementary Fig. 12). As the Ocr-bound structure only allows placement of a short, 20 bp, DNA molecule, it interacts with the C-terminal domain but does not enter the hinge region between N-terminal and C-terminal domains. We also demonstrated the requirement of charged residues in the PglX CTD to allow DNA binding and Ocr binding (Fig. 7). Our data should aid the

design of oligos for future structural studies of PglX bound to DNA, and supported efforts to engineer BREX activity (Fig. 6c).

Rational engineering of PglX broadened motif recognition, allowing the *Salmonella* BREX to target new phages and methylate new BREX motifs (Fig. 8). We were able to switch recognition for position -1 (relative to the point of methylation). MmeI recognises guanine at this position using R810 to form a hydrogen bond with guanine in the major groove, and an A774L mutant was shown to prevent binding of an A-T base pairing at position -1 through steric interference, switching specificity from R:Y to G:C[45,64]. The T802A and S838N mutations in PglX mutant 3 correspond to the positions of the A774 and R810 residues in MmeI, respectively, and are within the TRD. As rapid adaptability and evolution are vital factors in the phage-bacteria arms race that increase survivability of the local population[65], it follows that PglX would be the target of variability as a means to alter BREX defence specificity. Indeed, phase variation is common in *pglX* genes, but not other BREX components[10,66].

The inability of PglX to perform methylation during our in vitro reaction, nor when recombinantly expressed in the absence of other BREX genes in vivo[25], implies higher order BREX complexes are likely required. Such complexes could induce domain movements that would provide agreement with both proposed models of DNA binding. The arrangement of PglX monomers in the Ocr-bound structure is also potentially interesting, as a larger BREX complex might scan both sides of a dsDNA for the non-palindromic BREX motif by employing two PglX monomers, akin to the use by Type III and some dimeric Type II RM systems. Clearly, further work is needed on BREX components and complexes to uncover mechanistic details. The current study demonstrates that PglX contributes to BREX specificity, and it is involved in both the recognition and targeting of individual BREX motifs for host methylation and the resulting prevention of phage replication.

## Methods
### Bacterial strains
Strains used in this study are shown in Supplementary Table 2. We have described the *Salmonella* D23850Δφ strain previously[67]. The *Salmonella* D23850ΔφΔBREX and D23850ΔφΔ*brxL* strains were generated as described previously[39], using scarless lambda red recombination (Supplementary Fig. 2). Unless stated otherwise, *E. coli* strains DH5α (Invitrogen), BL21 (λDE3, Invitrogen) and ER2796 (NEB) were grown at 37 °C, either on agar plates or shaking at 220 rpm for liquid cultures. Luria broth (LB) was used as the standard growth media for liquid cultures, and was supplemented with 0.35% w/v or 1.5% w/v agar for semi-solid and solid agar plates, respectively. Growth was monitored using a spectrophotometer (WPA Biowave C08000) measuring optical density at 600 nm (OD$_{600}$). When necessary, growth media was supplemented with ampicillin (Ap, 100 μg/ml) or chloramphenicol (Cm, 25 μg/ml). Protein was expressed from pSAT1 or pBAD30 plasmid backbones by addition of 0.5 mM isopropyl-β-D-thiogalactopyranoside (IPTG) or 0.1% L-arabinose, respectively.

### Use of environmental phages
Phages used in this study are shown in Supplementary Table 2. Coliphages in the Durham phage collection have been described previously[33]. For *Salmonella* phages, sewage effluent was collected from a sampling site in Durham, courtesy of Northumbrian Water Ltd, as well as using samples from the River Wear. Filtrates were supplemented with 10 ml of LB, and inoculated with 10 ml of D23580ΔφBREX. Cultures were grown for 3 days before a 1 ml aliquots were transferred to sterile microcentrifuge tubes and centrifuged at 12000 × *g* for 5 min at 4 °C. The supernatants were transferred to new microcentrifuge tubes and 100 μl of chloroform was added to kill any remaining bacteria. Phage isolation was then carried out as previously described[33].

## Plasmid constructs and cloning

Primers used in this study are shown in Supplementary Table 3, and plasmids used in this study are shown in Supplementary Table 4. Ligation independent cloning (LIC) was utilised to create protein overexpression plasmids from pSAT1-LIC and pBAD30-LIC, as described previously[68]. This allowed the expression of fusion proteins with cleavable tags for efficient purification of recombinant proteins. The pBrxXL$_{Sty}$ plasmid was created previously[33] and contains the entire *Salmonella* D23580 BREX coding region, including the region 508 bp directly upstream of the *brxA* start codon to ensure that any promoters and transcriptional regulatory sites required for BREX expression and function were included. The creation of individual gene knockouts utilised Gibson Assembly (Gibson Assembly)[69]. Individual gene knockouts were designed within the context of the pBrxXL$_{Sty}$ vector to allow direct comparison on the same plasmid backbone. PCR primers were designed to amplify the pBrxXL$_{Sty}$ plasmid sequence either side of the gene to be removed (Supplementary Table 3). Primers were designed with overlapping regions to allow ligation of the amplicons via GA. GA designs consisted of 2-3 fragments of pBrxXL$_{Sty}$ produced by PCR with primers containing 20 bp homologous overlaps from upstream and downstream of the gene to be removed. Knockouts were designed for each of the six BREX genes, each of the two PARIS system genes, *ariA* and *ariB*, alongside an additional double knockout of both PARIS system genes. PCR-amplified and gel-purified fragments were pooled in an equimolar ratio to a final volume of 5 µl and added to 15 µl of assembly master mix. Reaction mixtures were incubated at 50 °C for 1 hr, then visualised on and gel purified from agarose gels. Resulting products which displayed the correct size were used to transform *E. coli* DH5α and cells were plated on Cm agar plates and incubated at 37 °C overnight. Plasmids from resulting colonies were extracted and sequenced (DBS Genomics) to confirm correct assembly. Gene knockouts for which GA was not successful were instead synthesised by Genscript. Primers for GA protocols were synthesised by IDT and were designed using the Benchling cloning design software, available online (benchling.com).

## Bioinformatics

To select bacterial genomes with a BREX system, Delta-BLAST was used to retrieve PglZ homologues[70]. Genomes encoding PglZ homologues (~17 K) were downloaded using the Entrez suite[71]. Hmmsearch from the HMMER suite, with custom hidden Markov models (HMM) for PglZ and PglX, was used to filter genomes that encoded both proteins (yielding ~12 K genomes)[72]. SearchIO and SeqIO from Biopython were subsequently employed to extract only genomic regions encompassing PglZ and PglX and any genes embedded within them[73]. Finally, PADLOC v2.0.0 and Defence-finder v1.2.0 were used to predict known defence systems encoded between PglZ and PglX[36,37]. All custom scripts used can be found at: https://github.com/GM110Z/Phage-defence-scripts.

## DNA sequencing

All genomic DNA extraction steps in this study were carried out using either a Zymo Miniprep Plus kit (Cambridge Biosciences) or a Monarch gDNA extraction kit (NEB). Bacterial genomic sequencing was performed by either MinION Mk1C nanopore sequencing or PacBio sequencing.

For MinION sequencing, DNA repair and end prep, barcode ligation and adaptor ligation steps were carried out according to Oxford Nanopore protocols (available at: community.nanopore.com) using the NEBNext Companion Module (New England Biolabs), Native Barcoding Expansions (EXP-NBD104 and EXP-NBD114) and ligation sequencing kit (SQK-LSK109), respectively. Sequencing was carried out using a MinION Flow cell (R9.4.1) on a MinION Mk1C. Following generation of raw sequencing data, basecalling was performed by the Guppy basecalling package (github.com/nanoporetech/pyguppy-client) either during sequencing or post sequencing and data was deconvoluted using the ont_fast5_api package (github.com/nanoporetech/ont_fast5_api). Megalodon was used for the detection of modified bases and the estimation of genomic methylation levels, with a 0.75 probability threshold for both modified and canonical bases for read selection and average percentage methylation calculations.

Libraries for sequencing were prepared using the SMRTbell Template Prep kit 3.0 (Pacific Biosciences). Bacterial gDNA was sheared using gTubes (Covaris) to produce DNA fragments with a mean size of 5–10 kb. The DNA was damage repaired and end repaired. SMRT-bell adaptors were then ligated. Exonuclease treatment removed Non SMRT-bell DNA. Sequencing was performed on a PacBio Sequel IIe (Pacific Biosciences). Data were analysed using PacBio SMRTAnalysis on SMRTLink_9.0 software Base Modification Analysis for Sequel data, to identify DNA modifications and their corresponding target motifs.

## Growth and infection curves

Phage growth and infection curves were carried out to monitor phage resistance conferred by pBrxXL$_{Sty}$ WT and pBrxXL$_{Sty}$ mutants in liquid culture. Growth was carried out in 200 µl culture volumes at 37 °C with shaking in a 96-well plate format, with $OD_{600}$ measurements taken every 5 min. Initial screening of inoculation and infection conditions produced optimal results with initial inoculation from overnight culture to $OD_{600}$ 0.1 and phage multiplicity of infection (MOI) of $10^{-6}$. As well as infection with phage TB34, a negative control – phage T7 – and a positive control (uninfected culture) were also run for each strain. All strains other than *E. coli* DH5α WT were grown with 25 µg/ml Cm.

## Efficiency of plating assays

Efficiency of plating (EOP) assays were carried out to assess the plaquing ability of phages in the Durham Phage Collection against *E. coli* DH5α pBrxXL$_{Sty}$ plasmids, and *Salmonella* phages against D23580 strains relative to controls. We used serial dilutions of high titre lysates in phage buffer and dilutions were mixed with overnight culture and molten 0.3% w/v agar, poured onto a 1% agar plate, dried and incubated overnight at 37 °C. For strains containing pBAD30 vectors, overnight cultures were induced with 0.2% w/v L-arabinose and incubated at 37 °C for 30 min prior to plating and both top and bottom agar layers included 0.2% w/v L-arabinose to induce continuous expression over the course of lawn growth. The EOP was calculated by dividing the pfu (plaque forming units) of the test strain by the pfu of the control strain. Data shown are the mean and the standard deviation of at least 3 biological and technical replicates.

## Protein expression and purification

All large-scale protein expression was performed in 1 L volumes of 2x YT broth in 2 L flasks with shaking at 180 rpm. In all cases, colonies from fresh transformation plates were used to inoculate 5;ml of 2x YT broth and grown overnight at 37 °C. This culture was then used to seed a 65 ml volume of 2x YT broth at 1:100 v/v and grown overnight at 37 °C to produce a second overnight culture. This culture was then used to seed 1 L of 2x TY at a 1:200 ratio, cultures were grown at 37 °C until exponential growth phase ($OD_{600}$ 0.3–0.7), induced at a final concentration of 0.5 mM IPTG, and protein was expressed at 18 °C overnight.

All purification steps were performed either on ice or at 4 °C. Fast protein liquid chromatography (FPLC) steps were carried out at 4 °C using an Akta Pure protein chromatography system (Cytiva). Protein purity was assessed using SDS-PAGE. Cells were harvested by centrifugation at 4000 rpm for 15 min at 4 °C and then resuspended in ice-cold A500 buffer (20 mM Tris HCl pH 7.9, 500 mM NaCl, 30 mM imidazole, 10% glycerol). Cells were lysed by sonication using a Vibracell VCX500 ultrasonicator, the soluble fraction was separated from insoluble cell material by centrifugation at $20000 \times g$ for 45 minutes at 4 °C and the supernatant was removed to a fresh, chilled tube for purification. Soluble cell lysate was applied to a 5 ml pre-packed Ni-NTA

His-Trap HP column (Cytiva) using a benchtop peristaltic pump at around 1.5 ml/min to allow binding of the 6xHis tag to the nickel resin. Columns were then washed with between 5–10 column volumes (CVs) of A500 to remove residual unbound protein and isocratic elution steps were performed using A500 buffer with imidazole concentrations adjusted to 30, 50, 90, 150 and 250 mM. Clean samples were pooled, dialysed into low salt A100 buffer (20 mM Tris HCl pH 7.9, 100 mM NaCl, 10 mM imidazole, 10% glycerol) and applied to a 5 ml HiTrap Heparin HP column (Cytiva), allowing separation of proteins with affinity for DNA. Bound protein was then washed with 5–10 CV of A100 and eluted using a salt gradient with C1000 buffer (20 mM Tris HCl pH 7.9, 1 M NaCl, 10% glycerol). Clean fractions were then pooled and digested with of human sentrin/SUMO-specific protease 2 (hSENP2) overnight at 4 °C to remove purification tags. Samples were then applied to a second Ni-NTA His-Trap HP column, this time allowing the now untagged protein of interest to flow through and removing remaining nickel binding contaminants. Successful tag cleavage and subsequent protein purity were assessed by SDS-PAGE, with tag cleavage visible as a noticeable reduction in protein molecular weight relative to tagged protein. Finally, size exclusion chromatography (SEC) was used to separate proteins by size, using a HiPrep 16/60 Sephacryl S-200 SEC column (Cytiva) connected to the FPLC system. Protein samples were dialysed overnight at 4 °C into S500 buffer (50 mM Tris HCl pH 7.9, 500 mM KCl, 10% glycerol) and concentrated to a 500 µl volume. The column was pre-equilibrated in S500, and the sample was loaded through a 500 µl volume capillary loop at 0.5 ml/min. The sample was eluted over 1.2 CVs at 0.5 ml/min and fractionated into 2 ml volumes for analysis by SDS-PAGE. Purified protein from SEC was concentrated to around 6 mg/ml and diluted in storage buffer (50 mM Tris HCl pH 7.9, 500 mM KCl, 70% glycerol) at a 1:2 ratio of protein to buffer, respectively, giving a final concentration of around 2 mg/ml. Samples were split into appropriately sized aliquots, snap-frozen in liquid nitrogen and stored at −80 °C for future use.

### Protein crystallisation and structure determination

Highly pure protein samples were used for crystallisation screening. Samples were either used immediately following purification or thawed on ice from −80 °C storage. Samples were dialysed into crystal buffer (20 mM Tris HCl pH 7.9, 150 mM NaCl, 2.5 mM DTT) and concentrated to 12 mg/ml. Protein concentration determination was performed using Nanodrop One (Thermofisher). Crystal screens were set using the sitting drop vapour diffusion method either by hand or using a Mosquito Xtal3 liquid handling robot (SPT Labtech). Crystal screens were incubated at 18 °C. All commercially available crystal screens were produced by Molecular Dimensions. For PglX and SAM samples, PglX was incubated with 1 mM SAM (Sigma) for 30 minutes on ice prior to addition to screens. For PglX-SAM:Ocr samples, PglX underwent the SAM incubation as above plus an additional 30 minute incubation on ice with 2.74 mg/ml of Ocr. Ocr was recombinantly expressed and purified as previously described[26,54]. PglX-SAM crystallised in 0.2 M potassium bromide, 0.1 M Tris pH 7.5, 8% w/v PEG 20000, 5% w/v PEG 500. PglX-SAM:Ocr crystallised in 0.1 M sodium/potassium phosphate pH 6.2, 14% w/v PEG 4000, 6% MPD. Crystallisation was confirmed by microscopy, with larger crystals extracted for X-ray diffraction. To harvest, 20 µl of screen condition was mixed with 20 µl of cryo buffer (25 mM Tris HCl pH 7.9, 187.5 mM NaCl, 3.125 mM DTT, 80% glycerol) and the solution was mixed thoroughly by vortexing. This solution was then added directly to the crystal drop at a 1 : 1 ratio. Crystals were extracted using nylon cryo loops and stored in liquid nitrogen until shipment. Data collection was carried out remotely at Diamond Light Source, Oxford, UK on beamlines I04 and I24, using their "Generic Data Acquisition" software (opengda.org).

Initial data processing was performed by automated processes on iSpyB (Diamond Light Source) using the Xia2-DIALS X-ray data processing and integration tool[74]. The same programme was used to merge multiple datasets and provide initial data on the space groups and unit cell sizes. Further data reduction and production of dataset statistics was carried out using AIMLESS within CCP4i2[75]. Merged datasets were first processed in CCP4i2 using BUCCANEER and REFMAC[75], and then iteratively built and refined in Coot[76] and Phenix[77], respectively. The quality of the final model was assessed using a combination of CCP4i2, Phenix, Coot and the wwPDB validation server. Visualisation and structural figure generation were performed in PyMol[57]. For PglX, the crystal structure was solved by molecular replacement in Phaser[78] using the PglX predicted model produced by AlphaFold[51]. The SAM molecule was downloaded from the PDB ligand repository and placed manually in Coot and similarly iteratively built and refined. The structure of the PglX-SAM:Ocr heterodimer complex was solved by molecular replacement in Phaser[78] using the PglX structure solved previously and the structure of Ocr (PDB 1S7Z).

### Analytical Size Exclusion Chromatography

Analytical SEC was performed on a Superdex 200 increase 10/300 GL SEC column (Cytiva) connected to an Akta Pure protein chromatography system (Cytiva). The column, system and loading loop were washed between each run and equilibrated with 1.2 CVs of A-SEC buffer (20 mM Tris-HCl pH 7.9, 150 mM NaCl). Protein samples were buffer exchanged into A-SEC buffer and concentrated. Final concentration ranged between 1 µM and 5 µM, as required to give a distinct measurable elution peak. Protein was loaded onto the system via a 100 µl capillary loop loaded using a 100 µl Hamilton syringe. For PglX-SAM:Ocr samples, PglX was incubated with each on ice in the same process as that used for crystallisation screening. Protein in capillary loops was injected into the column with 100 µl of A-SEC buffer and eluted over 1.2 CVs with A-SEC buffer at 0.375 ml/min. For estimation of protein molecular weight, relative to elution volume ($V_e$), a calibration curve was produced from commercially available high and low molecular weight protein calibration kits (Cytiva). Peaks were identified using the Unicorn 7 software package (Cytiva).

$V_e$ (elution volume) values were converted into the partitioning coefficient ($K_{av}$) for each sample using $V_c$ (geometric column volume) and $V_O$ (column void) in the equation:

$$K_{av} = \frac{V_e - V_o}{V_c - V_o}$$

The molecular weight calibration curve is then plotted as $K_{av}$ against $Log_{10}(M_r, kDa)$. The Stokes radius calibration curve plotted as $Log_{10}(R_{st}, Å)$ against $K_{av}$, allowing calculation of sample Stokes radius measurements. Estimated stokes radius calculations were carried out using the HullRad Stokes radius estimation server[53].

### Methyltransferase assay

SAM-dependant N6mA DNA methylation activity of PglX−alone and in equimolar combination with purified BrxB and PglZ−was probed in vitro using an MTase-Glo Methyltransferase Assay kit (Promega). The kit allows indirect measurement of SAM-dependent methyltransferase activity via production of the SAH reaction product. Through a proprietary two step reaction, SAH is used to produce ADP then ATP, which in turn is used by a luciferase reporter enzyme to generate a measurable luminescence signal. Signal can then be correlated to that produced by a SAH standard curve. The methyltransferase assay was carried out as per the manufacturer's instructions in a 96-well plate format. PglX, PglZ and BrxB were buffer exchanged into the methyltransferase assay reaction buffer (80 mM Tris pH 8.8, 200 mM NaCl, 4 mM EDTA, 12 mM MgCl2, 4 mM dithiothreitol (DTT)) and concentrated to 1 µM. As a substrate, 100 ng of E. coli DH5α genomic DNA was used per reaction as this should provide ample Salmonella BREX recognition motifs for methylation. The reaction mix was then combined with the protein samples at a 1:1 ratio with 10 µM of SAM and the

reaction was incubated at room temperature for 30 minutes. The SAH standard curve was prepared by two-fold serial dilutions of a 1 µM SAH stock in methyltransferase reaction buffer. Luminescence was measured on a Biotek Synergy 2 plate reader.

## Mass Photometry

Solution-phase mass determination of PglX species, with and without DNA or Ocr, was performed using the TwoMP (Refeyn) mass photometer. Samples were first diluted to 1 µM in low salt buffer A25 (10 mM Tris, pH 7.9; 25 mM NaCl; 0.5 µM SAM; 0.5 µM MgCl$_2$), and incubated for 1 h on ice, either alone or in combination with equimolar amounts of appropriate DNA oligos or Ocr. Samples were then further diluted in the same buffer to a final concentration of 5 nM and experimental data were obtained in the form of mass photometry videos recorded for one minute using the AcquireMP v2.5 software (Refeyn) on precleaned, poly-lysine-treated high sensitivity microscope slides. Mass calibration was done using thyroglobulin, aldolase and conalbumin from the same MW calibration kits (Cytiva) as used for analytical size exclusion chromatography. The experimental data were then fit to this calibration, and graphs were generated using the DiscoverMP v2.5 software (Refeyn).

## Reporting summary

Further information on research design is available in the Nature Portfolio Reporting Summary linked to this article.

## Data availability

The crystal structures of PglX-SAM and PglX-SAM:Ocr have been deposited in the Protein Data Bank under accession numbers 8C45 and 8Q56, respectively. All other data needed to evaluate the conclusions in the paper are present in the paper and/or Supplementary information. MinION and PacBio data that support the findings of this study have been deposited in the European Nucleotide Archive (ENA) at EMBL-EBI under accession number PRJEB71369. Source data are provided with this paper.

## Code availability

All custom scripts used can be found at: https://github.com/GM110Z/Phage-defence-scripts.

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

## Acknowledgements

We gratefully acknowledge Diamond Light Source for time on beamlines I04 and I24 under proposal MX24948. This work was supported by an

Engineering and Physical Sciences Research Council Molecular Sciences for Medicine Centre for Doctoral Training studentship [grant number EP/S022791/1] to S.C.W., a sLoLa grant from the Biotechnology and Biological Sciences Research Council [grant number BB/X003051/1] to S.C.W. and T.R.B., a Biotechnology and Biological Sciences Research Council Newcastle-Liverpool-Durham Doctoral Training Partnership studentship [grant number BB/M011186/1] to D.M.P., a responsive mode grant from the Biotechnology and Biological Sciences Research Council [grant number BB/Y003659/1] to T.R.B., and a Lister Institute Prize Fellowship to T.R.B. This work was supported in part by a Wellcome Trust Sir Henry Wellcome Fellowship [grant number 218622/Z/19/Z] to G.M., and a Wellcome Trust Senior Investigator award [grant number 106914/Z/15/Z] to J.C.D.H. For the purpose of open access, the authors have applied a CC BY public copyright licence to any Author Accepted Manuscript version arising from this submission.

## Author contributions

Analysed data: S.C.W., D.M.P., R.D.M., A.N., A.B., G.M., N.W. and T.R.B. Designed research: S.C.W., D.M.P., R.D.M., A.N., A.B., G.M., D.T.F.D., D.L.S., N.W., J.C.D.H. and T.R.B. Performed research: S.C.W., D.M.P., R.D.M., A.N., A.B., G.M., N.W. and T.R.B. Wrote the paper: S.C.W., D.M.P., A.N., G.M., D.T.F.D., J.C.D.H and T.R.B. Funding acquisition: J.C.D.H. and T.R.B. Supervised the study: D.L.S., J.C.D.H. and T.R.B.

## Competing interests

The authors declare no competing interests.
