## [Peer Review File · Nature Communications]

REVIEWER COMMENTS

Reviewer #1 (Remarks to the Author):

In this manuscript by Went et al., the authors characterized a BREX system from a pathogenic *Salmonella typhimurium* strain. They isolated environmental *Salmonella* phages against which the BREX system provided protection. Through individual gene deletion followed by phage infection assays in *E. coli*, the authors found that the BrxL subunit in this BREX system is dispensable for the antiphage function and in host methylation. The authors also determined the first structure for a PglX subunit in complex with a co-factor, SAM, providing structural details that shed light on this functionally important subunit involved in DNA methylation and sequence motif recognition. To understand how DNA binds to PglX, the authors further used a DNA-mimicking BREX inhibitor (Ocr) and employed comparative modeling to pinpoint the crucial region of methyltransferase for DNA binding. They attempted to identify the specificity determining region, which they further used for rational engineering and altering the target phage specificity.

Although the BREX operon included an antiphage PARIS system, the direct cooperation between these two systems couldn't be found, possibly limited by the phages screened in this study. While PARIS was shown to protect bacteria against T7 phages (in previous studies), no protection was observed in this specific PARIS system.

Overall, in my opinion, this study provides important information that advances the understanding of the BREX system and shed light on the structure of an essential subunit PglX and its mode of inhibition by Ocr. This study also present variability in BREX mode of action, especially considering the dispensability of BrxL in this specific case. The experiments are well-planned, and the claims are well supported by the presented data. However, I have a few major points, in addition to several minor points aimed at enhancing the clarity and readability of this manuscript, which I feel would be required before this manuscript is accepted for publication.

Major:

The authors have identified that the PARIS system is co-localized in this specific BREX system. Can the authors perform additional computational analyses to determine whether this co-localization is a single instance or if these two systems coexist more frequently in bacterial genomes?

Since the mechanism of BREX system-mediated antiphage protection is unclear, therefore, to explicitly state that BrxL is dispensable, the efficiency of plating assays for the BREX Del-BrxL construct should be performed in its native host as well.

Minor:

Line 61: The defense systems in Ref 9 were not identified through a guilt-by-association approach.

Line 263: “, though domains are missing,” can the author specify which domains they are referring to?

Line 275: 654-659 instead of 659-654.

Line 277: Please show the 2Fo-Fc electron density for the bound SAM in Fig 4.

Line 372: Can the authors show the specific negatively charged surface to which Ocr binds? I assume it should bind to a positively charged surface, given it is a highly negatively charged molecule.

Line 376: fix the typo.

Line 726: Include the concentration of the inducer used for protein expression.

Line 809: Define V_o , V_c , and V_e in this equation for clarity.

Figure S6: Specify in the figure legends that Mm1 is shown in "gray". It is also very difficult to see these superimpositions.

-Wherever DNA is modeled in figures, please specify that it's a "model" in the figure and legends.

Reviewer #2 (Remarks to the Author):

Went et al investigate the specificity of the BREX antiphage defense system from a clinically relevant strain of Salmonella and make progress towards understanding the molecular mechanisms underlying this system of phage defense. The major advances in this work include isolation of new Salmonella phages, determining the necessary BREX genes for phage defense in Salmonella, and determining the structure and specificity determinants of PglX as well as the structure of PglX bound to Ocr. This work builds on previous studies from the same laboratory examining BREX systems and undertakes the impressive task of 'reprogramming' PglX specificity through rational engineering of active site residues. Overall this paper is a strong contribution to the field and will be widely impactful to the phage defense community. I have several suggestions that could potentially improve the paper.

1. In figure 7b, no data is presented demonstrating that WT BREX does not methylate GATMAG motifs. The percent of GATMAG methylation with WT pBrxXLSty should also be calculated to demonstrate that mut3 does in fact actually have a new specificity compared to WT protein.
2. The authors claim that mutations enable re-programming of PglX. This is an overstatement, as only one out of 23 mutants was successfully reprogrammed. Are the other mutant forms of PglX that do not protect against Trib expressed in the cell? If so, can the authors explain why they fail to restrict Trib replication?
3. Relatedly, the authors demonstrate that phage Trib, without the targeted motif, is not susceptible to BREX defense but is when PglX is reprogrammed to recognize a motif that is present in the Trib genome. This suggests that BREX is acting directly on the phage DNA - if the authors incubate phage genomic DNA from TB34 and Trib with PglX and/or other BREX machinery is there any effect/damage to the phage DNA? Do any of the BREX proteins physically bind DNA? Although potentially outside the scope of this study, this would be a huge advance for the field.
4. The authors make great progress on understanding the mechanism of PglX methylation and specificity, but are unable to reconstitute methylation activity in vitro, indicating an incomplete understanding of the system. Advances in understanding the role of other BREX proteins in defense would greatly enhance

study - to this end, can the authors add other purified BREX components to try and reconstitute in vitro methylation activity? For example brxA, brxB, brxC, pglX, and pglZ appear to be required for methylation in vivo and likely play a role in vitro as well, as the authors point out in the text. Alternatively, perhaps AlphaFold predictions could support the formation of a higher order complex?

5. Sequencing and some level of characterization of the newly isolated Salmonella phages would also enhance the scope of the study and increase its impact for other researchers in the field. For instance, it would be interesting to determine whether the newly isolated phages encode an Ocr homolog. Sequencing susceptible vs. resistant phages may also help reveal patterns of phage defense provided by BREX.

6. Representative images of the raw data would be very helpful for interpretation of the plaque assays, and if it is possible, presentation of raw data from nanopore/pacbio sequencing may also help readers interpret/assess the data.

7. The crystal structure of PglX-Ocr is convincing evidence that the proteins form a heterotetrameric complex. However, the A-SEC traces are slightly confusing - although not required for publication SEC-MALS of these samples could be very informative for determining the identity of the unknown peaks.

8. The models of DNA-bound PglX are very informative. Are there specific mutations that could be used to determine which mode of DNA binding is required for phage defense? Similarly, the interacting residues between PglX and Ocr shown in Figure 5d should be mutated and tested for their role in complex formation.

9. It is unclear to me whether the "resulting motif" column in table S1 is data or a prediction? I believe it is a prediction but it would be more clear to re-label that column "predicted motif" or something similar.

I hope the authors will find my comments useful, thank you for the opportunity to read this interesting manuscript.

Reviewer #3 (Remarks to the Author):

Went et al. used plaque assays to confirm D23580 as a BREX/PARIS defense island in Salmonella with a particular phage target spectrum, and to show that it also protects against selected phages in an E. coli background. Via specific deletions, they showed that brxA, brxB, brxC, pglX and pglZ genes of the BREX system are essential for defense activity, while overexpression studies showed that the Ocr protein inhibited BREX defense activity. Using next-generation sequencing approaches, they identified a DNA motif as a target for N6mA methylation. BrxA, BrxB, BrxC, PglX and PglZ were required for full methylation activity. No activity was observed the putative methyltransferase; PglX, in an in vitro assay. The authors determined crystal structures of PglX and of PglX in complex with the Ocr inhibitor, Ocr. The structures confirmed homology of PglX to the MmI methyltransferase and suggested that Ocr might mimic, and thereby outcompete, substrate DNA on PglX. They exploited the structures for rational engineering of PglX to alter the methylation specificity.

The results presented are of interest to a community of researchers interested in phage defense, as they establish which elements of the BREX system are required for phage defense, and as the authors developed a toolbox that will allow further testing of mechanisms of phage defense. The results support

the hypothesis that PglX serves as a methyltransferase that is important for establishing specificity in phage defense by recognizing motifs targeted in phage and methylated in the host. The suggested mode of PglX inhibition by Ocr via DNA mimicry is interesting. The rational engineering approach has potential for reprogramming of the BREX phage spectrum. Most aspects of the work conducted appear technically sound. The manuscript is generally clearly written and should be accessible to a wide audience.

On the other hand, the mechanistic insights provided by the study at its present stage remain limited. Hypotheses derived from the structures have not been tested in a targeted manner. Lack of methyltransferase activity of isolated PgxL suggests that the picture of the active molecular methyltransferase machinery is presently incomplete.

Specific comments:

1. Abstract “This work identifies PglX as the sole specificity factor for methylation and phage defence within BREX” and Line 510 “PglX is the sole specificity factor ...” – These statements need to be tuned down. While the engineering data show that PglX is important for specificity, they do not exclude possible contributions from other factors, in particular in light of the authors’ failure to recapitulate PglX methyltransferase activity in vitro.
2. Line 411: “This implicates the C-terminal domain in DNA binding, ” and other statements referring to DNA binding by PglX – To confirm importance of regions for DNA binding and to support their model of DNA binding by PglX, the authors should conduct DNA binding studies with wildtype protein and suitable variants.
3. To validate the suggestion that Ocr acts as a DNA mimic, the authors should conduct DNA binding studies of PglX in the absence and presence of Ocr to show competition.
4. Could dimerization of PglX upon Ocr binding contribute to PglX inhibition and if so, how? If a potential role for dimerization in PglX inhibition can be discerned, it could be tested by rational mutagenesis and using the defense assays that the authors have established.
5. Line 349: “... leaving the 15.9 ml peak unidentified ...” and Figure S8: SDS PAGE analysis should be shown for the size exclusion chromatography runs. This would reveal if peaks correspond to protein or, e.g., contaminating nucleic acids. Also, absorbance at which wavelength is shown? Showing absorbance traces at 280 nm and 260 nm may also be revealing.
6. Line 266: ... assisting the solution and refinement of the crystallographic structure of Salmonella PglX bound to S-adenosyl-L-methionine (SAM), a co-factor for methylation, to 3.4 Å (Fig. 4; Table 1). Completeness, average intensity and CC1/2 for highest-resolution shell are very low, suggesting that effective resolution of the structure is lower than 3.4 Å. Rather than stating precise resolution claims, the authors should state that structures were refined by using all data collected up to 3.4 Å resolution or something equivalent. Also, SAM was inadequately modeled with hydrogen atoms.

7. For the PglX structure, the number reflections in the Rfree test set for the highest resolution shell is very small, suggesting that calculated Rfree values for the highest resolution shell are unreliable. Also, some values in Table 1 and in the validation report differ. Also, why is the B factor of the SAM ligand so much lower than the average B factor?

8. Line 384: “The exact orientation of ribose and methionine components of the molecule varied slightly, though this is likely due to variation in manual positioning of the molecule during refinement, as well as the resolution.” – Relevant regions of a omit or polder map and a comparison of SAM binding should be shown in the main text or supplement. What is meant by “variation in manual positioning of the molecule during refinement”?

9. Line 390: “Six salt bridges are produced between R79, N35, N42, N62, N76 and Q109 of Ocr and N1213, K1201, K1097, K1070, K1110, and K516 of PglX ...” – This cannot be seen in the Figure. Given the moderate resolution, how sure are the authors regarding side chain conformations, hydrogen bonds, salt bridges? Listed salt bridges cannot all be seen in the model provided. Also, how clear is density for solvent molecules modeled in the PglX-Ocr structure?

10. The authors should make diffraction data with phase angles available to reviewers to allow inspection of the qualities of the maps.

Minor points:

1. Figure 3 and Figure 7a: “>” signs for dark green wrong

2. Line 262: Comparison of AlphaFold model of PglX with Mmel appears repetitive with subsequent comparison with experimental structure. Rather state, after introducing the experimental structure, which features differ from the AlphaFold model. Also, domains missing in Mmel should be clearly described and illustrated.

3. Line 283: RMSD of 7.13 Å – for how many pairs of which type of atoms?

4. Figure S5a: Comparison is hard to discern, coloring is not explained in figure or legend. Instead of a superposition, a side-by-side comparison (or both) may be more effective.

5. Figure S5b: Which molecule is shown? Better to show/map functional domains in PglX?

6. Line 271: “However, the arrangement of the two copies allows only weak interactions that are likely formed due to interactions within the crystal rather than being biologically significant” – Perhaps refer to Figure S8a, showing that PglX is a monomer in solution.

7. Line 221: “Restoring active function of the Salmonella LT2 StySA BREX system identified GATCAG ...” – Please explain, how/why active function was/had to be restored. How was the motif actually identified?

8. Line 222, line 251 or Methods: References for MinION and PacBio sequencing are missing.

9. Figure 4: The binding mode of SAM in 4c is not well visible. Overall, the image shows little detail and is quite overloaded.

10. Table 1 – Entries should be trimmed to a reasonable number of decimals.

Ref: Nature Communications manuscript NCOMMS-23-62646

Title: Structure and rational engineering of the PglX methyltransferase and specificity factor for BREX phage defence

Response to Editor and Reviewers

We would like to thank the Editor and Reviewers for their positive comments and constructive criticism. Point-by-point responses to Editorial and Reviewer comments are listed below in blue, and where necessary, changes have been made to the original manuscript and are shown in red. In some instances, changes have been made to figures but these are not altered in colour, in order to maintain consistency within the figure.

Reviewer Comments

Reviewer #1 (Remarks to the Author):

In this manuscript by Went et al., the authors characterized a BREX system from a pathogenic *Salmonella typhimurium* strain. They isolated environmental *Salmonella* phages against which the BREX system provided protection. Through individual gene deletion followed by phage infection assays in *E. coli*, the authors found that the BrxL subunit in this BREX system is dispensable for the antiphage function and in host methylation. The authors also determined the first structure for a PglX subunit in complex with a co-factor, SAM, providing structural details that shed light on this functionally important subunit involved in DNA methylation and sequence motif recognition. To understand how DNA binds to PglX, the authors further used a DNA-mimicking BREX inhibitor (Ocr) and employed comparative modeling to pinpoint the crucial region of methyltransferase for DNA binding. They attempted to identify the specificity determining region, which they further used for rational engineering and altering the target phage specificity.

Although the BREX operon included an antiphage PARIS system, the direct cooperation between these two systems couldn't be found, possibly limited by the phages screened in this study. While PARIS was shown to protect bacteria against T7 phages (in previous studies), no protection was observed in this specific PARIS system.

Overall, in my opinion, this study provides important information that advances the understanding of the BREX system and shed light on the structure of an essential subunit PglX and its mode of inhibition by Ocr. This study also present variability in BREX mode of action, especially considering the dispensability of BrxL in this specific case. The experiments are well-planned, and the claims are well supported by the presented data. However, I have a few major points, in addition to several minor points aimed at enhancing the clarity and readability of this manuscript, which I feel would be required before this manuscript is accepted for publication.

Major:

The authors have identified that the PARIS system is co-localized in this specific BREX system. Can the authors perform additional computational analyses to determine whether this co-localization is a single instance or if these two systems coexist more frequently in bacterial genomes?

This is an excellent suggestion. We performed bioinformatic analyses that confirmed the frequency of not only BREX-PARIS associations, but the frequency of BREX associations with all known defence

systems as identified by DefenceFinder and PADLOC 2.0. The data are written at lines ~127-136 and presented as Fig. S1:

“As we have previously studied a phage defence island from *Escherichia fergusonii* ATCC 35469 plasmid pEFER that encodes a BREX system co-localised with a GmrSD-family Type IV restriction enzyme, BrxU ²², we were curious to determine how commonly BREX systems co-localise with other defence systems. DELTA-BLAST was first used to identify genomes encoding distant homologues of PglZ. From 12,000 genomes, ~9,000 genomes co-localised *pglZ* and *pglX* and ~7000 encoded a gene or genes between the BREX components. ~5,000 of these intergenic regions did not encode a currently known phage defence system in-between *pglZ* and *pglX*, as defined by either DefenseFinder ³⁷ or PADLOC 2.0 ³⁸. Of the remaining ~2000, 15.98% encoded BrxU homologues, whilst 2.21% encoded PARIS homologues, making them two of the most common defence systems associated with BREX (Fig. S1).”

Since the mechanism of BREX system-mediated antiphage protection is unclear, therefore, to explicitly state that BrxL is dispensable, the efficiency of plating assays for the BREX Del-BrxL construct should be performed in its native host as well.

This is another excellent suggestion, so we generated a *brxL* deletion in the original *Salmonella* D23580 host. We then set about testing our phages against this new *brxL* deletion strain. Unfortunately, when we came back to our *Salmonella* phage stocks we noted that they were isolated in 2019. Perhaps due to COVID interrupting lab activities, the phages had not been re-propagated for 4-5 years and we were unable to recover the specific phages described in the first submitted version of the manuscript. Undeterred, we went back out to the River Wear and sewage outlet in Durham, UK, from which the original phages had been isolated, to find new phages. We successfully isolated five new phages that we named SCW1-5.

Phages SCW1-5 were used to test EOP of both the Wild Type (WT) and *brxL* deletion strains, using the BREX deletion strain as control. The Wild Type EOP data are shown in Fig. 1b and the *brxL* deletion EOP data are shown in Fig. S4c, shown below for ease:

Phage	EOP WT/ Δ BREX	EOP Δ brxL / Δ BREX
SCW1	$3.33 \times 10^{-4} \pm 1.43 \times 10^{-4}$	$7.09 \times 10^{-4} \pm 1.16 \times 10^{-4}$
SCW2	0.93 ± 0.35	0.29 ± 0.08
SCW3	0.18 ± 0.09	0.16 ± 0.02
SCW4	1.15 ± 0.11	0.74 ± 0.26
SCW5	0.84 ± 0.23	1.10 ± 0.24

EOP $\geq 5 \times 10^{-1}$

$5 \times 10^{-1} > \text{EOP} \geq 10^{-1}$

$10^{-1} > \text{EOP} \geq 10^{-3}$

EOP $< 10^{-3}$

Phages SCW1 and SCW3 are BREX sensitive, and this is not impacted by *brxL* deletion. Phage SCW2 does not appear BREX sensitive when challenged with the WT strain, but we see enhanced BREX sensitivity with the *brxL* deletion. Phages SCW4 and SCW5 are insensitive to either strain. These data corroborate our previous data in *E. coli* (Fig. 3a), and demonstrate that (i) BrxL is dispensable in the native host and (ii) deletion of *brxL* can lead to an enhanced BREX activity.

The manuscript has been edited to reflect the use of new phages and these additional data with the *brxL* deletion strain.

Minor:

Line 61: The defense systems in Ref 9 were not identified through a guilt-by-association approach.

Apologies. We have fixed this to state, “functional selection”.

Line 263: “, though domains are missing,” can the author specify which domains they are referring to?

We have added in detail, “... though the N-terminal nuclease domain is missing from the Mmel structure.”

Line 275: 654-659 instead of 659-654.

Apologies, fixed.

Line 277: Please show the 2Fo-Fc electron density for the bound SAM in Fig 4.

This is a good idea and so instead of applying to only the PglX-SAM structure in Fig 4, we also wanted to include the PglX-SAM:Ocr structure in Fig. 5. We now show the 2Fo-Fc density maps for SAM molecules from both structures, as Fig. S10.

Line 372: Can the authors show the specific negatively charged surface to which Ocr binds? I assume it should bind to a positively charged surface, given it is a highly negatively charged molecule.

Correct – we have fixed the error to now say “positively”. Apologies!

Line 376: fix the typo.

We could not see a typo to fix, but we did change the sentence to “... perfectly align and abut one another”, to make it clearer.

Line 726: Include the concentration of the inducer used for protein expression.

Done

Line 809: Define V_o , V_c , and V_e in this equation for clarity.

Done

Figure S6: Specify in the figure legends that Mmel is shown in “gray”. It is also very difficult to see these superimpositions.

-Wherever DNA is modeled in figures, please specify that it's a "model" in the figure and legends.

We have worked to improve the superposition now shown as Fig. S6, and have indicated DNA is a model where required.

Reviewer #2 (Remarks to the Author):

Went et al investigate the specificity of the BREX antiphage defense system from a clinically relevant strain of Salmonella and make progress towards understanding the molecular mechanisms underlying this system of phage defense. The major advances in this work include isolation of new Salmonella phages, determining the necessary BREX genes for phage defense in Salmonella, and determining the structure and specificity determinants of PglX as well as the structure of PglX bound to Ocr. This work builds on previous studies from the same laboratory examining BREX systems and undertakes the impressive task of 'reprogramming' PglX specificity through rational engineering of active site residues. Overall this paper is a strong contribution to the field and will be widely impactful to the phage defense community. I have several suggestions that could potentially improve the paper.

1. In figure 7b, no data is presented demonstrating that WT BREX does not methylate GATMAG motifs. The percent of GATMAG methylation with WT pBrxXLSty should also be calculated to demonstrate that mut3 does in fact actually have a new specificity compared to WT protein.

Thank you for this suggestion. We have adjusted Fig. 8b (previously 7b) to show data for both GATCAG and GATAAG motifs for the WT system, which more clearly shows the change in specificity. The accompanying text has been changed to fit. Note we could not deconvolute the GATMAG for the mutant 3 BREX system into GATCAG and GATAAG, as the PacBio software provided data directly as GATMAG. Nevertheless, having the WT data shown as both does clearly indicate a change in specificity.

2. The authors claim that mutations enable re-programming of PglX. This is an overstatement, as only one out of 23 mutants was successfully reprogrammed. Are the other mutant forms of PglX that do not protect against Trib expressed in the cell? If so, can the authors explain why they fail to restrict Trib replication?

With great respect, we disagree. Though the success rate was low, demonstrating the difficulty of the task, we did nevertheless successfully re-program PglX. As such, we do not consider ourselves to have overstated the outcome. The result does after all demonstrate for the first time that it is possible to engineer PglX, and pinpoints PglX as the sole factor determining both the target for methylation, and the range of phages against which BREX can act. Previous work on re-programming Mmel (structurally related to PglX) was greatly improved after having obtained structural information from a DNA-bound complex (Callahan et al., 2016; <https://doi.org/10.1371/journal.pbio.1002442>). Without a DNA-bound structure for PglX, our success rate was lower. The unsuccessful mutants likely did not work as there was unproductive mutation of the PglX-DNA interaction. We have added a qualifying clause to the opening paragraph of the Discussion:

"Finally, though not a trivial task based on the number of mutants tested, we demonstrate successful rational engineering of BREX..."

3. Relatedly, the authors demonstrate that phage Trib, without the targeted motif, is not susceptible to BREX defense but is when PglX is reprogrammed to recognize a motif that is present in the Trib genome. This suggests that BREX is acting directly on the phage DNA - if the authors incubate phage

genomic DNA from TB34 and Trib with PglX and/or other BREX machinery is there any effect/damage to the phage DNA? Do any of the BREX proteins physically bind DNA? Although potentially outside the scope of this study, this would be a huge advance for the field.

We think this is an excellent idea. We have now performed and include mass photometry experiments (**Fig. 7**) examining the ability of PglX WT, the PglX NTD, and a multi-site mutant of PglX (PglX^{Mutant}), to bind both DNA and Ocr. This has demonstrated that charged residues of the CTD are essential for binding both DNA and Ocr.

The reviewer has also captured what we (and competitor labs in the field) are working towards: a model for BREX defence based on how the components interact to perform methylation, and then discriminate to stop replication of non-methylated DNA. There have indeed been multiple studies that investigate DNA interactions of BREX proteins, and our additional data included in revision adds to this effort. The original discovery paper demonstrates how phage DNA is not degraded by BREX, but is prevented from replicating (Goldfarb et al., 2015; <https://doi.org/10.15252/embj.201489455>). We then published a paper showing that BrxA can bind DNA (Beck et al., 2022; <https://doi.org/10.1016/j.crstbi.2022.06.001>). The Stoddard/Kaiser labs then characterised BrxL interacting with DNA (Shen et al., 2023; <https://doi.org/10.1093/nar/gkad083>). Since the current manuscript was submitted another study has been pre-printed, which tests all six core BREX components for DNA binding using EMSAs (Drobiazko et al., 2024; <https://doi.org/10.1101/2024.04.12.589305>). This study supports the previous work, indicating that BrxA, BrxL and indeed PglX bind DNA. Furthermore, it shows that the presence of a BREX motif improves PglX binding, and presents a PglX-DNA bound cryo-EM structure. It is worth pointing out that the senior author on the current manuscript is also an author on Drobiazko et al., 2024. For context, the current study and the Drobiazko et al. pre-print are outputs of an ongoing collaboration but due to the outbreak of the Russian-Ukraine war, we have not been able to publish these two studies together. This is specifically due to restrictions on data usage by Russian scientists when the data have been collected using certain UK government facilities such as Diamond Light Source.

4. The authors make great progress on understanding the mechanism of PglX methylation and specificity, but are unable to reconstitute methylation activity in vitro, indicating an incomplete understanding of the system. Advances in understanding the role of other BREX proteins in defense would greatly enhance study - to this end, can the authors add other purified BREX components to try and reconstitute in vitro methylation activity? For example brxA, brxB, brxC, pglX, and pglZ appear to be required for methylation in vivo and likely play a role in vitro as well, as the authors point out in the text. Alternatively, perhaps AlphaFold predictions could support the formation of a higher order complex?

We produced PglZ and BrxB and tested them in combination with PglX, but unfortunately we still observed no methylation activity. The updated data have been added to Fig. S7. Our next study will focus on pull-downs that map interactions between the BREX components, such that we can then test for methylation activity using either purified complexes, or complexes re-constituted from individually purified BREX proteins. The text has been updated (lines 316-319):

“No methylation was apparent from PglX under these conditions (**Fig. S7**), nor when we added in purified PglZ and BrxB. We hypothesize that PglX methyltransferase activity likely requires the presence of other BREX components, but the combination and ratio remains to be optimized.”

5. Sequencing and some level of characterization of the newly isolated *Salmonella* phages would also enhance the scope of the study and increase its impact for other researchers in the field. For instance, it would be interesting to determine whether the newly isolated phages encode an Ocr homolog. Sequencing susceptible vs. resistant phages may also help reveal patterns of phage defense provided by BREX.

We refer the reviewer to our recent paper by Kelly, Went et al., <https://doi.org/10.1128/aem.00623-23>), wherein we present characterisation and the genomes of 12 coliphages we previously isolated in order for use in BREX studies. We therefore fully support the utility of sequencing and characterising our isolated phages, but to do so for the new *Salmonella* phages, and potential “escape” resistant mutants thereof, is and of itself another publication beyond the remit of the current study. As prompted by the reviewer, the genomes have been isolated and are currently undergoing sequencing to support future work.

6. Representative images of the raw data would be very helpful for interpretation of the plaque assays, and if it is possible, presentation of raw data from nanopore/pacbio sequencing may also help readers interpret/assess the data.

We now provide an image of plates showing the changes in plaque morphology (and reduction of titre) for an example phage, Alma (Fig. S4b). The raw data for the sequencing are available in the designated repository (European Nucleotide Archive (ENA) at EMBL-EBI under accession number PRJEB71369). As raw data are not immediately easy to assess, we have also updated our graphic analysis in Fig. S5 to improve the ease of interpretation.

7. The crystal structure of PglX-Ocr is convincing evidence that the proteins form a heterotetrameric complex. However, the A-SEC traces are slightly confusing - although not required for publication SEC-MALS of these samples could be very informative for determining the identity of the unknown peaks.

Agreed, with apologies. We have repeated the analytical SEC on a new column and the read-out is much clearer. This is now presented as Fig. S9.

8. The models of DNA-bound PglX are very informative. Are there specific mutations that could be used to determine which mode of DNA binding is required for phage defense? Similarly, the interacting residues between PglX and Ocr shown in Figure 5d should be mutated and tested for their role in complex formation.

This is an excellent suggestion, and has been done. We have now performed and include mass photometry experiments (Fig. 7) examining DNA and Ocr binding by PglX WT, the PglX NTD, and a multi-site mutant of PglX (PglX^{Mutant}) that contains mutations of the residues identified in Fig. 5d. This has demonstrated that charged residues of the CTD are essential for binding both DNA and Ocr, supporting our models.

9. It is unclear to me whether the "resulting motif" column in table S1 is data or a prediction? I believe it is a prediction but it would be more clear to re-label that column "predicted motif" or

something similar.

We have re-labelled as suggested, thank you.

I hope the authors will find my comments useful, thank you for the opportunity to read this interesting manuscript.

We are glad the reviewer found the work interesting, and appreciate the useful comments that have enhanced the revised manuscript.

Reviewer #3 (Remarks to the Author):

Went et al. used plaque assays to confirm D23580 as a BREX/PARIS defense island in Salmonella with a particular phage target spectrum, and to show that it also protects against selected phages in an E. coli background. Via specific deletions, they showed that brxA, brxB, brxC, pglX and pglZ genes of the BREX system are essential for defense activity, while overexpression studies showed that the Ocr protein inhibited BREX defense activity. Using next-generation sequencing approaches, they identified a DNA motif as a target for N6mA methylation. BrxA, BrxB, BrxC, PglX and PglZ were required for full methylation activity. No activity was observed the putative methyltransferase; PglX, in an in vitro assay. The authors determined crystal structures of PglX and of PglX in complex with the Ocr inhibitor, Ocr. The structures confirmed homology of PglX to the Mm1 methyltransferase and suggested that Ocr might mimic, and thereby outcompete, substrate DNA on PglX. They exploited the structures for rational engineering of PglX to alter the methylation specificity.

The results presented are of interest to a community of researchers interested in phage defense, as they establish which elements of the BREX system are required for phage defense, and as the authors developed a toolbox that will allow further testing of mechanisms of phage defense. The results support the hypothesis that PglX serves as a methyltransferase that is important for establishing specificity in phage defense by recognizing motifs targeted in phage and methylated in the host. The suggested mode of PglX inhibition by Ocr via DNA mimicry is interesting. The rational engineering approach has potential for reprogramming of the BREX phage spectrum. Most aspects of the work conducted appear technically sound. The manuscript is generally clearly written and should be accessible to a wide audience.

On the other hand, the mechanistic insights provided by the study at its present stage remain limited. Hypotheses derived from the structures have not been tested in a targeted manner. Lack of methyltransferase activity of isolated PglX suggests that the picture of the active molecular methyltransferase machinery is presently incomplete.

Specific comments:

1. Abstract "This work identifies PglX as the sole specificity factor for methylation and phage defence within BREX" and Line 510 "PglX is the sole specificity factor ..." – These statements need to be tuned down. While the engineering data show that PglX is important for specificity, they do not

exclude possible contributions from other factors, in particular in light of the authors' failure to recapitulate PglX methyltransferase activity in vitro.

We have tuned down the language as requested. The abstract now reads:

“Our data demonstrate that PglX is used to recognise specific DNA sequences for BREX activity, providing motif recognition for both phage defence and host methylation.”

The discussion now reads:

“The current study demonstrates that PglX determines BREX specificity, and is responsible for both the recognition and targeting of individual BREX motifs for host methylation and the resulting prevention of phage replication.”

Beyond the two sections noted by the reviewer, we have also edited out two further instances of “sole specificity factor”.

Finally, please also note that in the last paragraph of the discussion we do comment that other BREX components are likely required.

2. Line 411: “This implicates the C-terminal domain in DNA binding, ” and other statements referring to DNA binding by PglX – To confirm importance of regions for DNA binding and to support their model of DNA binding by PglX, the authors should conduct DNA binding studies with wildtype protein and suitable variants.

This is an excellent suggestion and has been done, also as per request of Reviewer 2. We have now performed and include mass photometry experiments (**Fig. 7**) examining DNA and Ocr binding by PglX WT, the PglX NTD, and a multi-site mutant of PglX (PglX^{Mutant}) that contains mutations of the residues identified in **Fig. 5d**. This has demonstrated that charged residues of the CTD are essential for binding both DNA and Ocr, supporting our models.

3. To validate the suggestion that Ocr acts as a DNA mimic, the authors should conduct DNA binding studies of PglX in the absence and presence of Ocr to show competition.

There is an abundance of evidence in the literature that demonstrates how Ocr functions as a DNA mimic: Walkinshaw et al., 2002 ([10.1016/S1097-2765\(02\)00435-5](https://doi.org/10.1016/S1097-2765(02)00435-5)); Robert et al., 2012 (<https://doi.org/10.1093/nar/gks516>); Kanwar et al., 2016 (<https://doi.org/10.1093/nar/gkw234>), and others.

In addition, please see Drobiazko et al., 2024 (<https://doi.org/10.1101/2024.04.12.589305>) that contains the proposed Ocr competition assays, and the comment above as to the provenance of this pre-printed study on which the PI of our manuscript is an author (comment in response to Reviewer 2). We recognise the usefulness of these assays and so have included a sentence on Ocr competition and provided a reference to this pre-print within the text, at line 323.

4. Could dimerization of PglX upon Ocr binding contribute to PglX inhibition and if so, how? If a potential role for dimerization in PglX inhibition can be discerned, it could be tested by rational mutagenesis and using the defense assays that the authors have established.

PglX does not dimerize, as there is no evidence of large interacting surfaces between PglX protomers in either structure. This is backed up by computational analysis of interfaces using EMBL PISA, as we have now clarified within the results, lines 392-393:

“EMBL PISA⁵⁵ analysis of the complex showed that there was no dimerization of PglX in the complex, and identified important residues for Ocr interactions.”

5. Line 349: “... leaving the 15.9 ml peak unidentified ...” and Figure S8: SDS PAGE analysis should be shown for the size exclusion chromatography runs. This would reveal if peaks correspond to protein or, e.g., contaminating nucleic acids. Also, absorbance at which wavelength is shown? Showing absorbance traces at 280 nm and 260 nm may also be revealing.

We have repeated the analytical SEC on a new column and the read-out is much clearer and fits expected sizes. This is now presented within Fig. S9. We only have a single wavelength detector so the trace is at 280 nm, and this has now been stated in the figure legend.

6. Line 266: ... assisting the solution and refinement of the crystallographic structure of Salmonella PglX bound to S-adenosyl-L-methionine (SAM), a co-factor for methylation, to 3.4 Å (Fig. 4; Table 1). Completeness, average intensity and CC1/2 for highest-resolution shell are very low, suggesting that effective resolution of the structure is lower than 3.4 Å. Rather than stating precise resolution claims, the authors should state that structures were refined by using all data collected up to 3.4 Å resolution or something equivalent. Also, SAM was inadequately modeled with hydrogen atoms.

We have gone back and re-refined the model, cutting it to 3.5 Å. This has been re-submitted to the PDB. As a result, the R values are improved and we thank the Reviewer for prompting us to go and make this positive change. We have adjusted the comment in the text so it now reads:

“The structure was refined using all data collected up to 3.5 Å”

SAM ligands were built using the CIF file provided by the PDB and do include the hydrogens, we have simply chosen not to show hydrogens on either ligands or proteins.

7. For the PglX structure, the number reflections in the Rfree test set for the highest resolution shell is very small, suggesting that calculated Rfree values for the highest resolution shell are unreliable. Also, some values in Table 1 and in the validation report differ. Also, why is the B factor of the SAM ligand so much lower than the average B factor?

Please see comment above. We have re-refined and cut the resolution back to 3.5 Å, which has provided much improved values. Thank you for prompting us to do so.

We have also matched the values in Table 1 with the validation report (our apologies for the oversight). The B factor for the SAM ligands was calculated within the validation report.

8. Line 384: “The exact orientation of ribose and methionine components of the molecule varied slightly, though this is likely due to variation in manual positioning of the molecule during refinement, as well as the resolution.” – Relevant regions of a omit or polder map and a comparison of SAM binding should be shown in the main text or supplement. What is meant by “variation in manual positioning of the molecule during refinement”?

Thank you for this suggestion. We have produced 2Fo-Fc maps for all three SAM ligands and provided them as Fig. S10. We have adjusted the manuscript to now read:

“Though all three modelled SAM ligands across the two structures fit into the same pocket within PglX, the exact orientation of ribose and methionine components of the SAM ligands varied, likely due to the resolution of our models (Fig. S10).”

9. Line 390: “Six salt bridges are produced between R79, N35, N42, N62, N76 and Q109 of Ocr and N1213, K1201, K1097, K1070, K1110, and K516 of PglX ...” – This cannot be seen in the Figure. Given the moderate resolution, how sure are the authors regarding side chain conformations, hydrogen bonds, salt bridges? Listed salt bridges cannot all be seen in the model provided. Also, how clear is density for solvent molecules modeled in the PglX-Ocr structure?

Apologies that this was not clear. Important residues were identified using the EMBL PISA server to analyse interacting surfaces within the complex. We have now stated so within the results section. We have also re-designed Fig. 5d to make it easier to interpret. Furthermore, we have made a multi-site mutant removing the identified residues, and demonstrated that the mutant can no longer bind DNA or Ocr (Fig. 7), thereby supporting our model.

10. The authors should make diffraction data with phase angles available to reviewers to allow inspection of the qualities of the maps.

The structural data that were deposited into and verified by the PDB were uploaded with the submission, to allow assessment of electron density maps.

Minor points:

1. Figure 3 and Figure 7a: “>” signs for dark green wrong

Our apologies, these have been corrected throughout.

2. Line 262: Comparison of AlphaFold model of PglX with Mmel appears repetitive with subsequent comparison with experimental structure. Rather state, after introducing the experimental structure, which features differ from the AlphaFold model. Also, domains missing in Mmel should be clearly described and illustrated.

We have removed the earlier comparison as suggested, and clarified later on at lines 279-283:

“The closest structural homologue for the solved PglX structure, as designated by the DALI server ⁴⁵, remains the Type III restriction-modification system, Mmel ⁴⁶ (PDB 5HR4; Z-score 20.3), though the N-terminal nuclease domain that is found in Mmel (and not PglX) is missing from the Mmel structure. As a result, whilst Mmel demonstrates both N6mA DNA methyltransferase and DNA restriction activity ⁴⁶ the Mmel structure only has 60.8% sequence coverage against PglX...”

3. Line 283: RMSD of 7.13 Å – for how many pairs of which type of atoms?

This is now done.

4. Figure S5a: Comparison is hard to discern, coloring is not explained in figure or legend. Instead of a superposition, a side-by-side comparison (or both) may be more effective.

We have worked to improve the comparison as requested, now shown as Fig. S6.

5. Figure S5b: Which molecule is shown? Better to show/map functional domains in PglX?

We have labelled this panel for clarity (now shown as Fig. S6b).

6. Line 271: “However, the arrangement of the two copies allows only weak interactions that are likely formed due to interactions within the crystal rather than being biologically significant” – Perhaps refer to Figure S8a, showing that PglX is a monomer in solution.

This would put our narrative out of order but please note we already call back to the earlier figure when we introduce the SEC data at line 346:

“These data indicate that PglX exists as a monomer in solution, supporting our conclusions from the PglX-SAM structure (Fig. 4).”

7. Line 221: “Restoring active function of the Salmonella LT2 StySA BREX system identified GATCAG ...” – Please explain, how/why active function was/had to be restored. How was the motif actually identified?

This refers to work done by a different group in a separate publication, not by us. To clarify, we have simplified the text:

“Study of the *Salmonella* LT2 StySA BREX system identified GATCAG as the target motif sequence³¹.”

8. Line 222, line 251 or Methods: References for MinION and PacBio sequencing are missing.

We have now added refs.

9. Figure 4: The binding mode of SAM in 4c is not well visible. Overall, the image shows little detail and is quite overloaded.

We have re-made Fig. 4c to help with interpretation. Thank you for prompting us, we think it now looks much clearer.

10. Table 1 – Entries should be trimmed to a reasonable number of decimals.

This has been done.

REVIEWER COMMENTS

Reviewer #1 (Remarks to the Author):

In the revised version of this manuscript by Went et al., the authors included data from several experiments which further support their claims. I found that while the new experiments were included, the text was not updated in the results section. Additionally, in my opinion, the quality (including accuracy) of this manuscript could be further improved.

Here are my suggestions:

While the authors did a great job analyzing the co-localization of BREX with other known defense systems, the presentation of this data in Fig. S1 is hard to understand in its current form. Either the authors should increase the size and update it with a better coloring scheme, or this data could be presented in a tabular form. In its current style, it is hard to comprehend.

The revised size exclusion chromatography (SEC) data are very clean, and the authors did a great job performing these experiments; however, the text was not updated. For example, the Ocr SEC profile has only one peak in this new data, but the text says “multiple species”; the 15.5 mL peak should be updated to 11.2 mL as per new data in Fig S9a; and so on... The hydrodynamic radius calculation should also be repeated with this revised data. Please check carefully; there is more that needs to be updated!

Lines 392-395: There is inconsistency between the residue names in the text and labels in Fig. 5d. For example, the text says N35, N42, N62, N76, and Q109, while in Fig. 5d these are labelled as D35, D42, D62, D76, and E109. Please double-check and fix these issues.

The mass photometry-based interaction data and oligomeric state(s) calculations with and without DNA are a great addition. However, the total number of counts used in the final analysis, the percentage population of each species, and the standard deviation of the Gaussian fit are not defined. These should be noted in the Figures or Figure legends. For PglX NTD and DNA interaction plots, is it possible that the observed peak is a mixture of non-interacting DNA and PglX NTD rather than just PglX NTD as shown in Fig. 7?

In Fig. S13b inset, there is a white rectangular box hiding the bound SAM ligand. Please remove that.

Please double-check the references. For example, citations 9 and 11 are repetitive.

Reviewer #2 (Remarks to the Author):

The authors have addressed my concerns and I believe the manuscript is greatly improved and suitable for publication.

Reviewer #3 (Remarks to the Author):

Please refer to the attached file.

Ref: Nature Communications manuscript NCOMMS-23-62646A

Title: Structure and rational engineering of the PglX methyltransferase and specificity factor for BREX phage defence

Response to Editor and Reviewers

We would like to again thank the Editor and Reviewers for their positive comments and constructive criticism. Point-by-point responses to Editorial and Reviewer comments are listed below in blue, and where necessary, changes have been made to the revised manuscript and are shown in red. In some instances, changes have been made to figures but these are not altered in colour, in order to maintain consistency within the figure. Our second round of revisions have been added on top of previous revisions within the manuscript.

Reviewer Comments

Reviewer #1 (Remarks to the Author):

In the revised version of this manuscript by Went et al., the authors included data from several experiments which further support their claims. I found that while the new experiments were included, the text was not updated in the results section. Additionally, in my opinion, the quality (including accuracy) of this manuscript could be further improved.

Our sincere apologies for missing some updates. We have addressed your points below.

Here are my suggestions:

While the authors did a great job analyzing the co-localization of BREX with other known defense systems, the presentation of this data in Fig. S1 is hard to understand in its current form. Either the authors should increase the size and update it with a better coloring scheme, or this data could be presented in a tabular form. In its current style, it is hard to comprehend.

Agreed, it was over-complex. We have converted the data to a simple table. Thank you for the suggestion.

The revised size exclusion chromatography (SEC) data are very clean, and the authors did a great job performing these experiments; however, the text was not updated. For example, the Ocr SEC profile has only one peak in this new data, but the text says “multiple species”; the 15.5 mL peak should be updated to 11.2 mL as per new data in Fig S9a; and so on... The hydrodynamic radius calculation should also be repeated with this revised data. Please check carefully; there is more that needs to be updated!

Sincere apologies on this oversight. We have fixed and updated this section, which now also has an additional SDS-PAGE gel of the peak fractions. The gel results confirm our predicted assignments for each peak (Fig. S9b). The new text is at lines 344-366, copied below for ease:

“We then aimed to recreate a PglX:Ocr complex²⁷ using our purified *Salmonella* PglX and Ocr, and visualise the resulting complexes with analytical SEC. Elution volume is dependent on protein

molecular weight, and can also reflect the shape and size of the protein molecule itself. The hydrodynamic radius of protein samples seen by analytical SEC can be calculated from the observed K_{av} value⁵³, allowing comparison to the calculated hydrodynamic radius of predicted protein and protein complex models produced by AlphaFold⁵⁴. The solution state of native PglX was determined using analytical SEC. PglX eluted from the SEC column at 11.2 ml (**Fig. S9a**), which indicated a hydrodynamic radius of 54.8 Å, matching the 57 Å calculated hydrodynamic radius of PglX. These data indicate that PglX exists as a monomer in solution, supporting our conclusions from the PglX-SAM structure (**Fig. 4**). The Ocr sample was then examined by analytical SEC in isolation (**Fig. S9a**). The Ocr SEC profile gave a single species at 14.6 ml, with a calculated hydrodynamic radius of 30.4 Å. Ocr is known to be a dimer in solution^{26,52}, which would be 27.6 kDa and corresponds to a calculated hydrodynamic radius of 24.3 Å. Purity of the Ocr sample was confirmed by mass spectrometry and SDS-PAGE (**Fig. S9b and c**). PglX and Ocr were then combined at a 1:2 molar ratio prior to SEC (**Fig. S9a and b**). The combined sample produced an additional peak at 10.3 ml beyond those from the individual PglX and Ocr samples (**Fig. S9a and b**), and moved the bulk of the PglX peak. The peak at 10.3 ml indicated a large complex of approximately ~391 kDa, potentially comprised of at least two copies of PglX, and Ocr dimers (**Fig. S9a**). A model of two monomers of PglX and one Ocr dimer produced by AlphaFold produced a predicted hydrodynamic radius of 58.3 Å, compared to a calculated hydrodynamic radius of 63.9 Å for the observed A-SEC peak. This suggested that the additional peak eluting at 10.3 ml represented a PglX-Ocr heterotetramer in solution. The protein contents of each peak were examined by SDS-PAGE, and the results matched our assignments, wherein PglX and Ocr could be found to have shifted and co-elute at 10.3 ml (**Fig. S9b**)."

Lines 392-395: There is inconsistency between the residue names in the text and labels in Fig. 5d. For example, the text says N35, N42, N62, N76, and Q109, while in Fig. 5d these are labelled as D35, D42, D62, D76, and E109. Please double-check and fix these issues.

Apologies again for the error. The correct residues are **D35, D42, D62, D76, and E109** and this has now been fixed in the manuscript.

The mass photometry-based interaction data and oligomeric state(s) calculations with and without DNA are a great addition. However, the total number of counts used in the final analysis, the percentage population of each species, and the standard deviation of the Gaussian fit are not defined. These should be noted in the Figures or Figure legends. For PglX NTD and DNA interaction plots, is it possible that the observed peak is a mixture of non-interacting DNA and PglX NTD rather than just PglX NTD as shown in Fig. 7?

Thank you, we were similarly pleased with the outcomes of these experiments. As requested, we tried multiple ways to add the back the data. Our original Refeyn figure (Fig. 7) was styled as per Figs 1 and 3 of this paper (coincidentally a *Nature Communications* paper: <https://www.nature.com/articles/s41467-022-29841-0>). Owing to the multiple peaks in our analysis, we found that including the quant data and the cartoons as guides generated a confusing and cluttered figure. To ensure easy interpretation by a reader we have chosen to keep Fig. 7 mostly unchanged, but have introduced a new Supplementary Figure, Fig. S14, that contains all of the data for each peak in full, plotted against number of counts. Interested readers can then go and investigate these data by referring to Fig. S14. We hope this is viewed as a reasonable compromise, given the restraints due to the number of peaks requiring labels.

As to the second point – the peak for PgIX NTD does not shift when DNA is added, indicating that there is no DNA interaction, as the Reviewer points out. The peak in experiments of PgIX NTD with DNA (of each form), could well represent a mixed peak. A line has therefore been added to the manuscript noting this possibility and the relevant DNA icon has been added to Fig. 7.

In Fig. S13b inset, there is a white rectangular box hiding the bound SAM ligand. Please remove that.

Done.

Please double-check the references. For example, citations 9 and 11 are repetitive.

References have been curated and corrected.

Reviewer #2 (Remarks to the Author):

The authors have addressed my concerns and I believe the manuscript is greatly improved and suitable for publication.

Thank you for your assessment.

Reviewer #3 (Remarks to the Author):

[copied verbatim below including colours from the Reviewer's pdf. Our responses are prefixed by >> and highlighted in yellow]

In revising their manuscript, the authors adequately addressed several issues raised by this reviewer. However, some of the original points remain.

>>Apologies for not satisfactorily addressing all your points. We believe we have now done so below.

Original statements/issues in black

Author replies in blue

New comments in red

1. Statements concerning the role of PglX as a specificity factor have not been tuned down adequately.

“Our data demonstrate that PglX is used to recognise specific DNA sequences for BREX activity, providing motif recognition for both phage defence and host methylation.”

It should read, e.g.:

“Our data demonstrate that PglX is used to recognise specific DNA sequences for BREX activity, contributing to motif recognition for both phage defence and host methylation.”

“The current study demonstrates that PglX determines BREX specificity, and is responsible for both the recognition and targeting of individual BREX motifs for host methylation and the resulting prevention of phage replication.”

It should read, e.g.:

“The current study demonstrates that PglX contributes to BREX specificity, and that it is involved in both the recognition and targeting of individual BREX motifs for host methylation and the resulting prevention of phage replication.”

>>We have adjusted as requested.

2. Concerning SEC analyses:

We have repeated the analytical SEC on a new column and the read-out is much clearer and fits expected sizes. This is now presented within Fig. S9. We only have a single wavelength detector so the

trace is at 280 nm, and this has now been stated in the figure legend.

I agree that the chromatograms now look cleaner. But I do not understand why the authors refuse to show SDS-PAGE analyses of peak fractions that would clearly identify which proteins make up the peaks.

>>Please understand that we were in no way refusing to do this work. We simply felt the data had sufficient clarity following improvements in the SEC runs due to use of a new column. However, as requested, we now provide an SDS-PAGE analysis (Fig. S9b) and refer to this new data in the main text. The SDS-PAGE analysis confirms our peak assignments.

3. Concerning quality of the diffraction data, Table 1:

We have gone back and re-refined the model, cutting it to 3.5 Å. This has been re-submitted to the PDB. As a result, the R values are improved and we thank the Reviewer for prompting us to go and make this positive change. We have adjusted the comment in the text so it now reads:

“The structure was refined using all data collected up to 3.5 Å”

In principle fine, but instead of showing improved statistics for highest resolution shells, the authors simply omit these values completely from modified Table 1. Contrary to their response, the overall statistics for the datasets did not change much; they are (and they previously were) fine. The highest resolution shell parameters should be quoted as had been done in the original manuscript.

>>We have added these data into Table 1.

4. Concerning modelling of SAM:

SAM ligands were built using the CIF file provided by the PDB and do include the hydrogens, we have simply chosen not to show hydrogens on either ligands or proteins.

I do not understand this response. Of course a geometry/parameter/model file obtained from the PDB or a program may include hydrogens. That does not mean you have to keep them in the model and refine them if data quality does not warrant refinement of hydrogens. At the present resolutions, hydrogen atoms should not be refined and should not be part of the models deposited in the PDB.

>>The CIF file we generated using eLBOW produced major errors in refinements when uploaded to the PDB servers, and so we used the CIF file from the PDB itself. We do not feel that inclusion of the ligand hydrogens is a major point as it does not impact our conclusions, nor the model build at this resolution, and the model was accepted by the PDB. We politely request that we agree to disagree on this issue.

5. Relevant regions of a omit or polder map ... should be shown in the main text or supplement.

Thank you for this suggestion. We have produced 2Fo-Fc maps for all three SAM ligands and provided them as Fig. S10.

2Fo-Fc maps can be model biased. As initially suggested, in my view the authors should provide omit or polder maps.

>>We have replaced all of the maps in Fig. S10 with Polder maps, as suggested. Thank you for raising this issue – we have not used Polder before and the outputs worked very nicely. We will add this to our general practice.

REVIEWERS' COMMENTS

Reviewer #1 (Remarks to the Author):

The authors have now fixed the errors and addressed all of my concerns. In my opinion, this revised version is a strong candidate for publication in Nature Communications. Therefore, I recommend the publication of this study.

Thank you for giving me the chance to read this.

Reviewer #3 (Remarks to the Author):

In revising their manuscript, the authors adequately addressed the points raised by this reviewer.

Ref: Nature Communications manuscript NCOMMS-23-62646B

Title: Structure and rational engineering of the PglX methyltransferase and specificity factor for BREX phage defence

Response to Editor and Reviewers

We would like to again thank the Editor and Reviewers for their positive comments and constructive criticism. Point-by-point responses to Editorial and Reviewer comments are listed below in blue, and where necessary, changes have been made to the revised manuscript and are shown in red. In some instances, changes have been made to figures but these are not altered in colour, in order to maintain consistency within the figure. Our second round of revisions have been added on top of previous revisions within the manuscript.

Reviewer Comments

Reviewer #1 (Remarks to the Author):

The authors have now fixed the errors and addressed all of my concerns. In my opinion, this revised version is a strong candidate for publication in Nature Communications. Therefore, I recommend the publication of this study.

Thank you for giving me the chance to read this.

Thank you.

Reviewer #2 (Remarks to the Author):

[no additional comments provided]

Reviewer #3 (Remarks to the Author):

In revising their manuscript, the authors adequately addressed the points raised by this reviewer.

Thank you.